# Sleep cycle-dependent vascular dynamics in male mice and the predicted effects on perivascular cerebrospinal fluid flow and solute transport

Laura Bojarskaite [1,2,6] ✉, Alexandra Vallet[3,6], Daniel M. Bjørnstad[1,6], Kristin M. Gullestad Binder[1], Céline Cunen [3,4], Kjell Heuser[2], Miroslav Kuchta[5], Kent-Andre Mardal[3,5] & Rune Enger [1] ✉

Perivascular spaces are important highways for fluid and solute transport in the brain enabling efficient waste clearance during sleep. However, the underlying mechanisms augmenting perivascular flow in sleep are unknown. Using two-photon imaging of naturally sleeping male mice we demonstrate sleep cycle-dependent vascular dynamics of pial arteries and penetrating arterioles: slow, large-amplitude oscillations in NREM sleep, a vasodilation in REM sleep, and a vasoconstriction upon awakening at the end of a sleep cycle and microarousals in NREM and intermediate sleep. These vascular dynamics are mirrored by changes in the size of the perivascular spaces of the penetrating arterioles: slow fluctuations in NREM sleep, reduction in REM sleep and an enlargement upon awakening after REM sleep and during microarousals in NREM and intermediate sleep. By biomechanical modeling we demonstrate that these sleep cycle-dependent perivascular dynamics likely enhance fluid flow and solute transport in perivascular spaces to levels comparable to cardiac pulsation-driven oscillations.

Perivascular spaces (PVS) lined by the astrocytic endfeet, are key passageways for movement and exchange of fluids and solutes, and play important roles for drug delivery into the brain and waste clearance[1,2]. Removal of extracellular waste products like β-amyloid (Aβ) is crucial for brain health and the prevention of neurodegenerative diseases such as Alzheimer's disease (AD)[3]. Already in the 1970s pioneering work of Helen Cserr and colleagues identified the perivascular compartments of the brain as pathways enabling bulk flow and efficient transport of solutes out of the brain[4]. A model for brain waste clearance—the glymphatic system as proposed in a seminal study ten years ago[1]—states that cerebrospinal fluid (CSF) flows along pial arteries, enters the brain via PVS of penetrating arterioles, then flows through the parenchyma collecting extracellular waste, before it exits in PVS along veins[5]. Other studies also found evidence for clearance along arteries[6]. From there, waste may exit the brain along cranial and spinal nerves, via arachnoid granulations (although recently debated)[7] and meningeal lymphatic vessels[8,9], which all drain into the cervical lymphatic vasculature[10]. In addition, there might exist a bidirectional CSF and solute flow along the arteries[6]. Some of the mechanistic underpinnings of the glymphatic system, such as driving

[1]GliaLab and the Letten Centre, Division of Anatomy, Department of Molecular Medicine, Institute of Basic Medical Sciences, University of Oslo, 0317 Oslo, Norway. [2]Department of Neurology, Oslo University Hospital, 0027 Oslo, Norway. [3]Department of Mathematics, University of Oslo, 0316 Oslo, Norway. [4]Norwegian Computing Center, 0314 Oslo, Norway. [5]Department of Numerical Analysis and Scientific Computing, Simula Research Laboratory, Kristian Augusts gate 23, 0134 Oslo, Norway. [6]These authors contributed equally: Laura Bojarskaite, Alexandra Vallet, Daniel M. Bjørnstad. ✉e-mail: laura.bojarskaite@medisin.uio.no; rune.enger@medisin.uio.no

forces for perivascular and parenchymal flow and exit pathways, are still debated[11]. Even so, there appears to be a consensus that the PVS serve as highways for efficient transport of extracellular fluid and solutes in the brain.

Flow along the PVS is thought to be facilitated by mechanical forces created by the vasculature[12,13]. Specifically, heartbeat-driven pial artery pulsations have been demonstrated to propel fluorescent microspheres along vessels at the brain surface[13]. Moreover, pharmacologically manipulating blood pressure and heart frequency seems to affect clearance of waste products from the brain[13]. Recent studies have also shown that vasomotion of longer time scales observed in wakefulness may play a role in propelling CSF[14,15].

Brain waste clearance has been shown to be considerably more active in sleep[3,16]. This was first demonstrated in 2013 in naturally sleeping head-fixed mice[16]. This link between sleep and brain waste clearance has later been reinforced by studies in humans showing that sleep deprivation is associated with reduced clearance and increased build-up of harmful proteins such as Aβ in the brain, and increased risk of neurodegenerative diseases[3]. The mechanisms underlying the enhancement of waste clearance during sleep are not well understood but have been proposed to depend on an increased extracellular space[16] and coupled blood-CSF flow patterns in non-rapid eye movement (NREM) sleep[17]. Recently, sleep state coupled changes in brain perfusion and blood flow dynamics of large surface vessels have been demonstrated in mice[18,19]. Yet the importance of a complete sleep cycle, including NREM sleep, intermediate state (IS)[20], REM sleep, microarousals and awakening after each sleep cycle[21] on vascular dynamics and its consequences for PVS dynamics, has not been demonstrated.

Given that CSF fluxes in the brain are dependent on vascular dynamics and that there are prominent blood flow changes in sleep, we hypothesized that sleep specific vasomotion could govern the size of the PVS and facilitate flow of CSF. Hence, we set out to measure the changes in the PVS across the different sleep-wake states by two-photon microscopy. We show that blood vessels exhibit sleep state dependent dynamics. Specifically, we observed slow, large amplitude oscillations of both pial and penetrating arterioles in slow-wave sleep, coupled to reciprocal changes to the PVS of penetrating arterioles, followed by large arterial and arteriolar dilations that started during IS sleep and reached maximum dilation during REM sleep and lasted for the entire duration of REM episode, coupled to a near obliteration of the PVS. During the brief arousal at the end of a sleep cycle after REM sleep and microarousals in NREM and IS sleep, arteries and arterioles constrict and PVS enlarges. Using biomechanical modeling we demonstrate that these sleep cycle-dependent PVS dynamics, and in particular slow vasomotion in NREM sleep, likely play a salient role in driving fluid flow and solute transport in the PVS.

## Results

### Two-photon imaging of vascular dynamics in natural sleep

To assess the vascular dynamics throughout the sleep cycle, we performed two-photon microscopy linescans across blood vessels in the somatosensory cortex of naturally sleeping *GLT1*-eGFP transgenic mice expressing enhanced green fluorescent protein (eGFP) in astrocytes with the vasculature outlined by Texas Red-labeled dextran (Figs. 1a, b and 2a, b)[22]. Pial arteries and veins with corresponding penetrating vessels were identified by their morphology and direction of blood flow. We classified sleep-wake states using an infrared sensitive camera, electrocorticography (ECoG) and electromyography (EMG) (Supplementary Fig. 1). We identified three different behavioral states of wakefulness by analyzing mouse movements on the IR video footage (Supplementary Fig. 1a): locomotion, spontaneous whisking, and quiet wakefulness. In mice, locomotion is tightly associated with whisking[23], and for this reason our locomotion behavioral state comprises both movement and whisking. Using standard criteria on ECoG

and EMG[20,24] signals, we identified three sleep states: NREM sleep, IS sleep, and REM sleep (Supplementary Fig. 1b). NREM sleep and IS sleep are sub-states of SWS, where IS sleep is a transitional state from NREM to REM sleep, characterized by an increase in sigma (10–16 Hz) and theta (5–9 Hz) power, and a decrease in delta (0.5–4 Hz) ECoG power[20,22]. The mice were trained to fall asleep without any use of anesthesia or sedatives, as established in Bojarskaite et al. 2020, enabling us to monitor a natural progression of sleep states[22].

### Pial arteries display sleep-cycle dependent vascular dynamics

We first assessed pial artery vascular dynamics by measuring vessel lumen diameter using two-photon microscopy *x-t* linescans (Fig. 1b). We observed striking changes in the diameter of pial arteries across the different sleep states and wakefulness (Fig. 1c). NREM and IS sleep were associated with very low frequency (VLF 0.1–0.3 Hz) and low frequency (LF 0.3–1 Hz) oscillations in the vessel diameter (Fig. 1d, e and Supplementary Table 1). VLF and LF oscillations of comparable amplitudes to NREM and IS sleep were only observed during locomotion across all sleep-wake states (Supplementary Fig. 2 and Supplementary Table 2). Upon REM sleep, a pronounced dilation of the arteries ensued that even outmatched the dilation observed upon locomotion (Fig. 1f, Supplementary Fig. 2, and Supplementary Tables 1–2). Subsequently, upon brief awakenings at the end of a sleep cycle after a REM episode[21], pial arteries constricted to reach the diameter observed during quiet wakefulness (Fig. 1f and Supplementary Table 1). These data clearly show that every part of the sleep cycle entails unique pial artery vascular dynamics and is in line with previous reports on hemodynamic aspects of sleep[19].

### Size of penetrating arteriole PVS changes across sleep cycle

Next, we measured vascular dynamics of penetrating arterioles (Fig. 2). Vessel lumen and endoot tube diameters were measured in *x-t* line scans in the red and green channels, respectively, and the total width of the PVS was assessed as the difference between the vessel lumen diameter and the endoot tube diameter (Fig. 2b). We defined PVS as the void between the lumen and endoot sleeve to encompass all the potential pathways of CSF and solute flow in two-photon image recordings[11]. However, defining PVS in this way will over-estimate the actual PVS volume, as there are cellular and non-cellular constituents within this compartment, such as smooth muscle cells, macrophages, fibroblasts, and extracellular matrix proteins[25,26], that likely hinder fluid flow. For example, the wall of penetrating arterioles consists of smooth muscle cells that would contract upon vasoconstriction. We do not expect the total volume of the smooth muscle cells to change during contraction, and similarly, we do not expect the volume of the other cellular- and non-cellular components of the PVS to change with vessel diameter changes. For this reason, the over-estimation of the PVS volume should be constant across the different sleep states and vascular diameters.

Similar to pial arteries, penetrating arterioles displayed prominent changes in the diameter of the vessel lumen (Fig. 2c). Interestingly, we also observed changes in the diameter of the endoot tube and PVS total width (Fig. 2c). NREM and IS sleep were associated with VLF and LF oscillations in the arteriole lumen and endoot tube diameter that were mirrored in PVS size changes (Fig. 2d, Supplementary Figs. 3a, 4, 5b, c, and Supplementary Tables 3–6). These slow VLF and LF oscillations were considerably less prominent in all other sleep and wake states, except for locomotion (Supplementary Fig. 6a, b, d and Supplementary Tables 7–10). When the mice entered REM sleep, the VLF and LF oscillations diminished and arteriole lumen and endoot tube dilated, while PVS size shrunk (Fig. 2e, Supplementary Figs. 3b, 5a, and Supplementary Tables 3–6). Strikingly, the magnitude of REM associated vasodilation and PVS shrinkage was even larger than what was observed during locomotion (Supplementary Fig. 6c and Supplementary Tables 7–10). In contrast, upon the brief awakening at the end

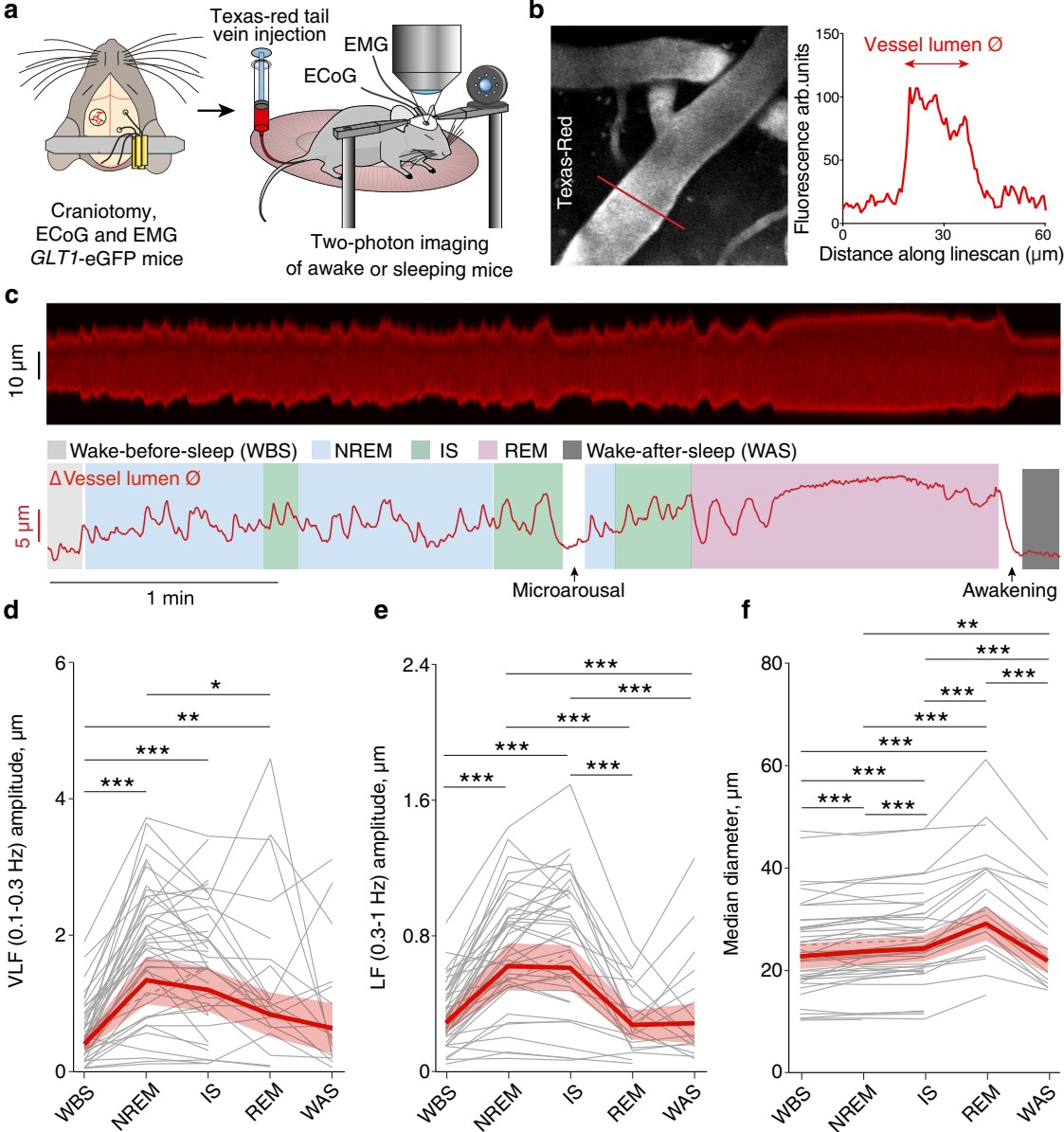

**Fig. 1 | NREM, IS and REM sleep states are associated with specific pial artery diameter changes. a** Experimental setup. *GLT1*-eGFP transgenic mice were fitted with a cranial window exposing the somatosensory cortex, ECoG electrodes and EMG electrodes. The vasculature was outlined with Texas Red-labeled dextran. **b** (*Left*) Representative image of a pial artery with Texas-Red labeled dextran in the lumen. Imaging was performed by placing cross sectional *x-t* line scans. Linescan length 60 μm. (*Right*) Vessel lumen diameter (Ø) was determined along a line across the vessel. Experiment was repeated on 44 pial arteries from 5 mice. **c** Representative traces of a pial artery line scan and change in vessel lumen diameter (Ø) during a sleep cycle. **d** Amplitude of very low frequency (VLF 0.1–0.3 Hz), **e** amplitude of low frequency (LF 0.3–1 Hz) oscillations and **f** median diameter of pial artery lumen throughout a sleep cycle. Every gray line (dashed or full)

represents an individual vessel, dashed lines are used when an observation is missing in a certain state for a given vessel (for example dashed line between WBS and IS means that we do not have an observation for that particular vessel in NREM state), bold lines and shaded area are the estimates and 95% CI from linear mixed effects models (two sided) **d**: *n* = 570, in 44 vessels, and 5 mice, **e, f**: *n* = 579 episodes, 44 vessels, 5 mice. *$P < 0.05$, **$P < 0.01$, ***$P < 0.001$, Tukey adjustment for multiple comparisons. See supplementary information for exact p-values, and degrees of freedom for the statistical tests. WBS wake before sleep; NREM non-rapid eye movement sleep; IS intermediate state sleep; REM rapid eye movement sleep; WAS wake after sleep; ECoG electrocorticography; EMG electromyography. Source data are provided as a Source Data file.

of a sleep cycle after a REM episode and during microarousals in NREM and IS sleep, arteriole lumen and the endfoot tube constricted to reach a similar size as in quiet wakefulness, while the PVS enlarged (Fig. 2e, Supplementary Figs. 3b, c, 5a and Supplementary Tables 3–6). Interestingly, the PVS was larger during wake-after-sleep than during wake-before-sleep (Fig. 2e, Supplementary Figs. 5a, and Supplementary Tables 4, 5). Such NREM, IS and REM associated vascular dynamics were not detected in venules (Supplementary Fig. 7). Sleep cycle-dependent vascular dynamics were confirmed in 2D time-series

imaging experiments (Supplementary Fig. 8 and Supplementary Movie 1). These results show that every sleep cycle state is associated with specific PVS dynamics.

**NREM/IS slow vasomotion increases predicted PVS fluid velocity**
To assess the effects of sleep-state-specific VLF and LF oscillations on fluid flow in PVS, we performed biomechanical modeling based on individual vessel measurements (Fig. 3a). The simulations assumed a fluid-filled annular PVS of length 600 μm with a uniform cross-

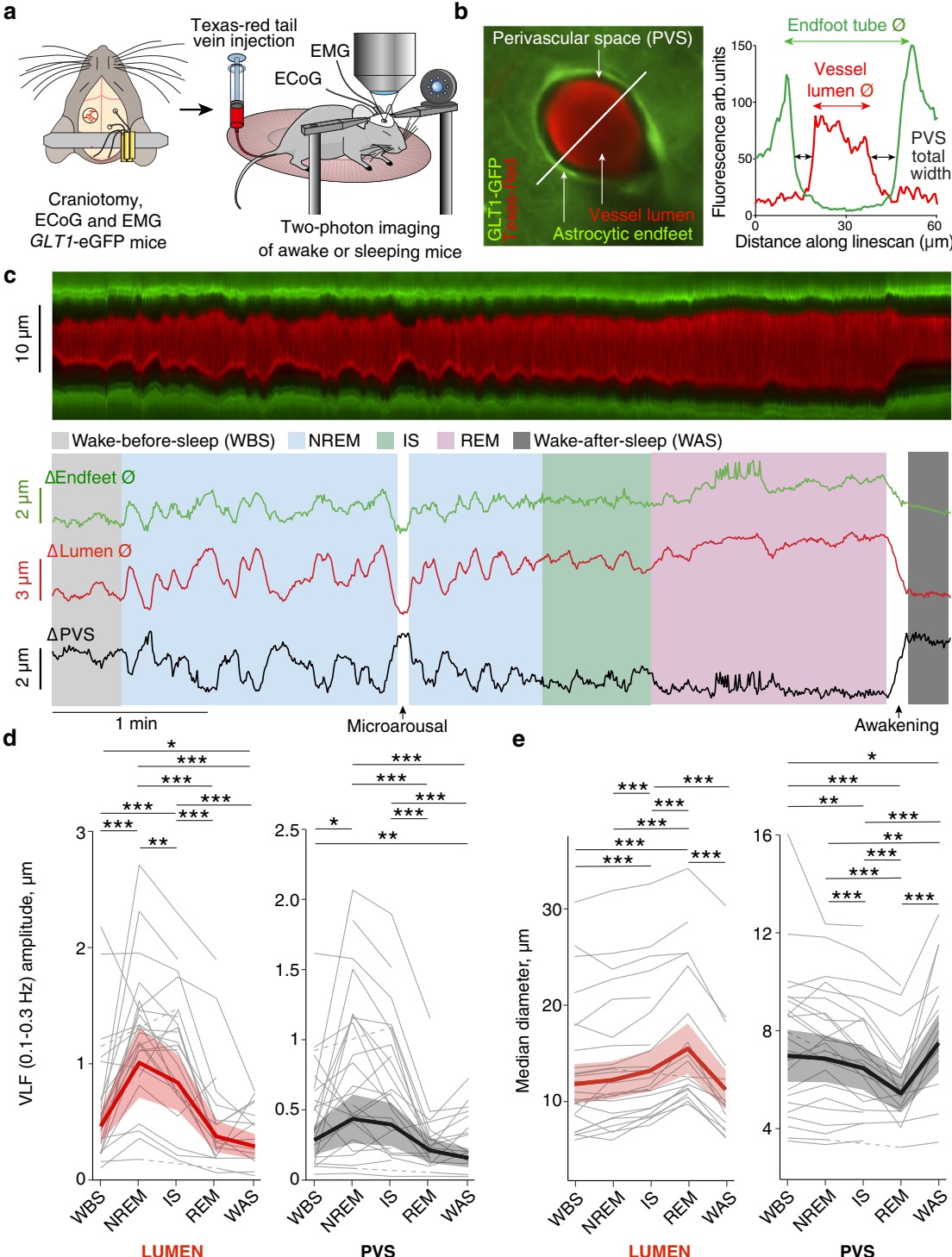

**Fig. 2 | NREM, IS and REM sleep states are associated with specific penetrating arteriole diameter changes that are mirrored in the size of the PVS.**
**a** Experimental setup. *GLT1*-eGFP transgenic mice were fitted with a cranial window exposing the somatosensory cortex, ECoG electrodes and EMG electrodes. The vasculature was outlined with Texas Red-labeled dextran. **b** (*Left*) Representative image of a penetrating arteriole with *GLT1*-eGFP fluorescence in astrocytic endfeet and Texas-Red labeled dextran in the vessel lumen. Imaging was performed by placing cross sectional *x-t* line scans. (*Right*) Vessel lumen diameter (Ø) and astrocyte endfoot tube diameter (Ø) was determined along a line across the vessel. Total width of the PVS was assessed as the difference between the vessel lumen diameter and the endfoot tube diameter. **c** Representative traces of a penetrating arteriole line scan, change in endfoot tube diameter (Ø), vessel lumen diameter (Ø) and perivascular space diameter during a sleep cycle. **d** Amplitude of very low

frequency (VLF 0.1–0.3 Hz) oscillations and **e** median diameter of lumen and total width of PVS throughout a sleep cycle. Every gray line (dashed or full) represents an individual vessel, dashed lines are used when an observation is missing in a certain state for a given vessel (for example dashed line between WBS and IS means that we do not have an observation for that particular vessel in NREM state), bold lines and shaded area are the estimates and 95% CI from linear mixed effects models (two-sided), $n = 310$ episodes, 25 penetrating arterioles, 4 mice. *$P < 0.05$, **$P < 0.01$, ***$P < 0.001$, Tukey adjustment for multiple comparisons. See supplementary information for exact *p*-values, and degrees of freedom for the statistical tests. WBS wake before sleep; NREM non-rapid eye movement sleep; IS intermediate state sleep; REM rapid eye movement sleep; WAS wake after sleep; ECoG electro-corticography; EMG electromyography; PVS perivascular space. Source data are provided as a Source Data file.

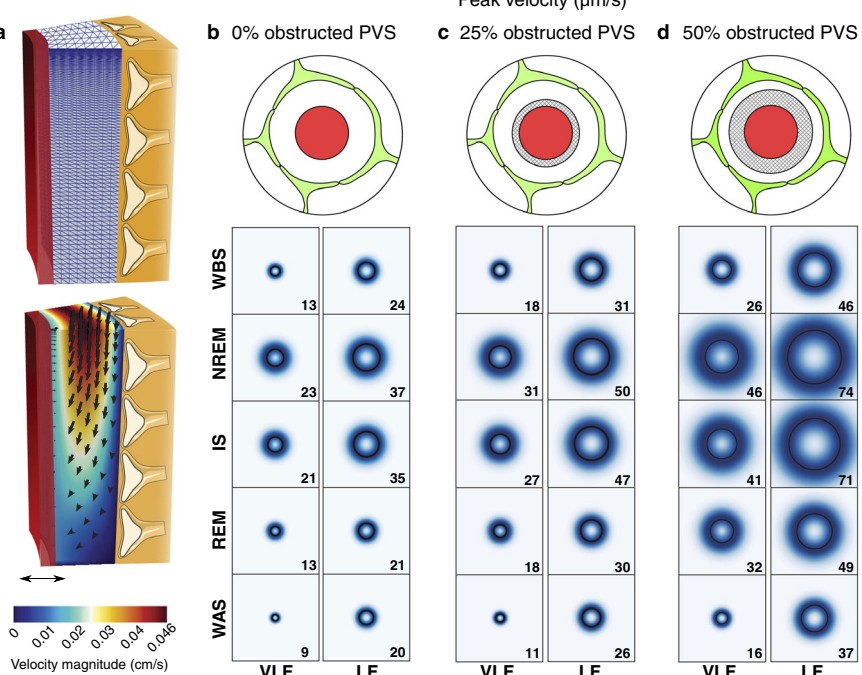

**Fig. 3 | NREM/IS slow vasomotion increases predicted PVS fluid velocity.**
**a** Illustration showing the model of the PVS (top), and representative vasomotion-driven CSF flow simulation (bottom). **b–d** Peak fluid velocity in penetrating arteriole PVS generated by VLF and LF oscillations during a sleep cycle as predicted by biomechanical modeling using three scenarios where we added a fixed volume to the PVS obstructing flow corresponding to 0, 25, and 50% of our measured PVS in sectional deformation during vasomotion all along the vessel segment.

quiet wakefulness. The black circle represents the median, whereas the shading represents the distribution of all modeled vessels (*n* = 16 vessels, 4 mice). The value of the median is also given in the lower right corner for each box. WBS wake before sleep; NREM non-rapid eye movement sleep; IS intermediate state sleep; REM rapid eye movement sleep; WAS wake after sleep; PVS perivascular space; VLF very low frequency; LF low frequency. Source data are provided as a Source Data file.

The model does not enforce net pressure differences or net flow in the simulations. We found that the VLF and LF oscillations in NREM and IS increased CSF peak velocities compared to the other sleep-wake states (Fig. 3b and Supplementary Fig. 9a). As there are cellular and non-cellular constituents within PVS, such as smooth muscle cells, macrophages, fibroblasts, and extracellular matrix proteins that hinder fluid flow, we also simulated scenarios where we added a fixed volume to the PVS obstructing flow corresponding to 25 and 50% of our measured PVS in quiet wakefulness (Fig. 3c, d and Supplementary Fig. 9b, c). We found that a smaller fraction of free PVS resulted in higher CSF peak velocities in all sleep-wake states (Fig. 3b–d and Supplementary Fig. 9a–c). Importantly, NREM and IS slow oscillation-driven CSF peak velocities as predicted by our modeling (0% obstructed PVS: 21–37 μm/s, 25%: 27–50 μm/s, 50%: 41–74 μm/s) were on the same order of magnitude as the bulk CSF velocities measured by imaging fluorescent microbeads moving alongside pial arteries in mice[13] (18.7 μm/s, 95% confidence interval 9.4–28 μm/s). These simulations predict a likely salient role of NREM and IS sleep specific slow vasomotion as a driving force for fluid flow in the PVS in addition to cardiac vessel oscillations.

**NREM/IS slow vasomotion enhances predicted dispersion in PVS**
The effect of the resultant oscillating flow on solute transport was then modeled by considering the spread of 2000 and 70 kDa dextran tracers in the PVS (Fig. 4 and Supplementary Figs. 10 and 11). In the model, the instantaneous flow will enhance the spread of the molecule compared to diffusion due to dispersion (Fig. 4a). We modeled this enhanced tracer spread using the amplitudes of the observed VLF and LF oscillatory patterns in the PVS size for every vessel observed through the different sleep stages. Compared to diffusion, VLF and LF oscillations enhanced tracer spread with the largest effect observed in NREM sleep (Fig. 4b and Supplementary Figs. 10a and 11a). In wakefulness (wake-

before-sleep state), we estimate the tracer spread to be on average 11.8 and 3.1% larger than pure diffusion for 2000 and 70 kDa, respectively (Fig. 4b, Supplementary Fig. 10a). In contrast, the tracer spread due to LF and VLF flow during NREM state is considerably larger, on average 29.1 and 6.7% larger than diffusion for 2000 and 70 kDa, respectively (Fig. 4b and Supplementary Figs. 10a). In addition, the enhancement of tracer spread due to LF and VLF oscillations varies more in NREM sleep than in quiet wakefulness and can reach maximum values up to 140 and 30% during NREM sleep compared to maximum values of 35 and 6% in quiet wakefulness for 2000 kDa and 70 kDa tracer, respectively (Supplementary Figs. 10a and 11a). As with the CSF flow simulations we also simulated scenarios where we added a fixed volume to the PVS obstructing flow corresponding to 25 and 50% of our measured PVS in quiet wakefulness. We found that a smaller fraction of free PVS resulted in a higher enhancement factor in all sleep-wake states for both 2000 and 70 kDa solutes (Fig. 4c, d and Supplementary Figs. 10b, c and 11b, c). The enhancement factor for tracer dispersion in vein PVS was insignificant compared to arteriole PVS and reached maximum values of 5.3 and 1.6% for 2000 and 70 kDa tracers, respectively (Supplementary Fig. 12).

Previous modeling studies have predicted that cardiac oscillations lead to a 100% enhancement of the transport of 70 kDa dextran by dispersion and peak oscillatory CSF flow velocity of 100–150 μm/s[27–29]. However, these studies deduced the PVS cross-sectional area change from the vessel wall movement only, assuming the astrocyte endfoot tube as rigid, which is not the case in our recordings (Fig. 2 and Supplementary Fig. 3). The cardiac driven CSF peak velocities, when astrocyte deformation is taken into account, are expected to be much lower, on the order of 25–50 μm/s for a 250 μm long PVS[27]. In the present study, we were not able to assess the oscillatory size changes of the PVS of penetrating arterioles in the cardiac frequencies, as the diameter changes were close to the imaging resolution limit, and that we could not rule out an effect of rigid movements of the vessel on

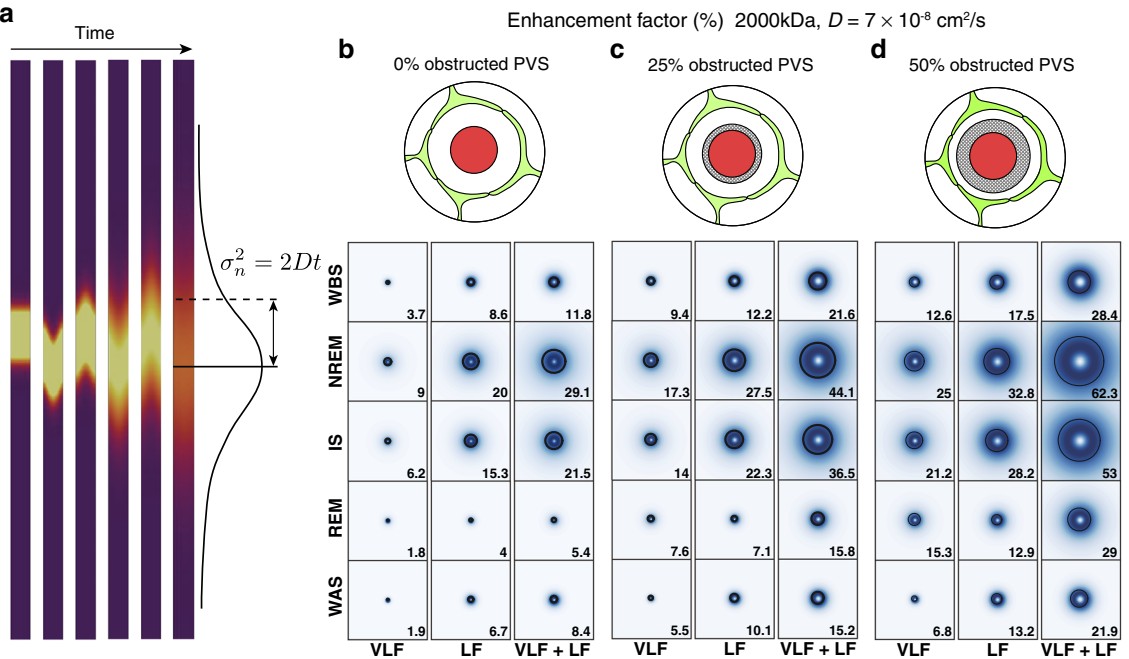

**Fig. 4 | NREM/IS slow vasomotion enhances predicted dispersion in PVS.**
**a** Modeling of the slow vasomotion driven tracer spread in the PVS.
**b**–**d** Enhancement factor for 2000 kDa tracer spread in penetrating arteriole PVS generated by VLF and LF oscillations during a sleep cycle as predicted by biomechanical modeling using three scenarios where we added a fixed volume to the PVS obstructing flow corresponding to 0, 25, and 50% of our measured PVS in quiet wakefulness. The enhancement factor is the relative increase in solute movement induced by VLF and LF oscillatory flow, compared to pure diffusion. The black circle represents the median, whereas the shading represents the distribution of all modeled vessels (*n* = 16 vessels, 4 mice). The value of the median is also given in the lower right corner for each box. WBS wake before sleep; NREM non-rapid eye movement sleep; IS intermediate state sleep; REM rapid eye movement sleep; WAS wake after sleep; PVS perivascular space; VLF very low frequency; LF low frequency. Source data are provided as a Source Data file.

these small measured amplitudes. Therefore, we modeled three different scenarios with cardiac oscillations driving CSF fluid peak velocities at 10, 50, and 100 μm/s, and imposed the associated PVS cross section area change (Supplementary Figs. 13 and 14). The dispersion enhancement factor for such cardiac velocities is then of the same order of magnitude as to what we find for the VLF and LF oscillations during NREM sleep (for 0% obstructed PVS: LF/VLF – 6.7 and 29.1% while 50 μm/s cardiac – 5.4 and 13.6%, for 70 and 2000 kDa, respectively) (Fig. 4b and Supplementary Figs. 10 and 13). Overall, our simulations predict that VLF and LF slow oscillations that are of largest amplitude during NREM sleep enhance the dispersion of solutes within the arterial PVS to levels comparable to cardiac-driven oscillations.

### NREM slow vasomotion enhances predicted solute influx
We next analyzed how VLF and LF oscillations affected solute movement into the PVS of a penetrating arteriole, a process relevant not only to the glymphatic system, but also to drug delivery into the brain. We assumed a fluid filled volume with a solute concentration of 1 at the brain surface entrance of the penetrating arteriole PVS, and simulated how the oscillatory vessel movements moved the solute concentration front (defined as the point with a concentration of 0.1) as it traversed the depth of the PVS. VLF and LF oscillations considerably enhanced the movement of 70 and 2000 kDa solutes into the PVS compared to pure diffusion in both wakefulness and sleep and, as expected, the solutes moved faster in NREM sleep compared to quiet wakefulness (Fig. 5a). For example, in quiet wakefulness, it took 86 s for the concentration front of 70 kDa dextran to spread to 100 μm of depth compared to only 68 s in NREM sleep. The same transport by pure diffusion without any oscillatory flow would have taken 112 s. As with the CSF flow and dispersion simulations we also performed solute influx simulations in which 25 and 50% of PVS area was obstructing fluid flow and solute transport, as is likely the case due to the cellular and non-cellular constituents within PVS, such as smooth muscle cells,

macrophages, fibroblasts, and extracellular matrix proteins that hinder fluid flow. We found that a smaller fraction of free PVS resulted in faster solute influx (Fig. 5b, c). For example, in NREM sleep it took 68 s for the concentration front of 70 kDa dextran to spread to 100 μm of depth in the case of 0% obstructed PVS, while it took 56 s in the case of 25% obstructed PVS and only 36 s in the case of 50% obstructed PVS. To conclude, our modeling data predict that VLF and LF oscillations increase solute influx in the arteriolar PVS compared to pure diffusion and that the enhancement of solute influx is greatest during NREM sleep when VLF and LF are of largest amplitude.

## Discussion
The mechanisms underlying fluid dynamics and solute transport in the brain's PVS during sleep remain elusive, partly because of the lack of data on the entire sleep cycle, including the natural progression of NREM sleep, IS sleep, REM sleep, the awakening at the end of the sleep cycle and microarousals in NREM and IS sleep. Using naturally sleeping head-fixed mice we here show that each state of the sleep cycle displays unique arteriole diameter changes that are coupled to changes in the size of PVS. Using biomechanical modeling we predict that slow, large-amplitude oscillatory vasomotor patterns in NREM sleep are able to generate oscillatory fluid movement on a similar magnitude to heartbeat-generated fluid movement in the PVS and enhance solute transport in PVS. This supports the hypothesis that blood volume and CSF dynamics are coupled and that vasomotion could be a main driver for CSF bulk flow in the PVS together with heart-beat driven oscillations[13,14,17]. It is important to stress that these observations from mathematical models have to be confirmed with biological experiments, such as for instance labeling CSF and PVS with fluorescent tracers. The next steps will be to understand how REM sleep-specific reduction in the size of PVS and the subsequent enlargement of PVS upon awakening at the end of the sleep cycle or during the microarousals in NREM and IS sleep affect fluid flow and solute transport in

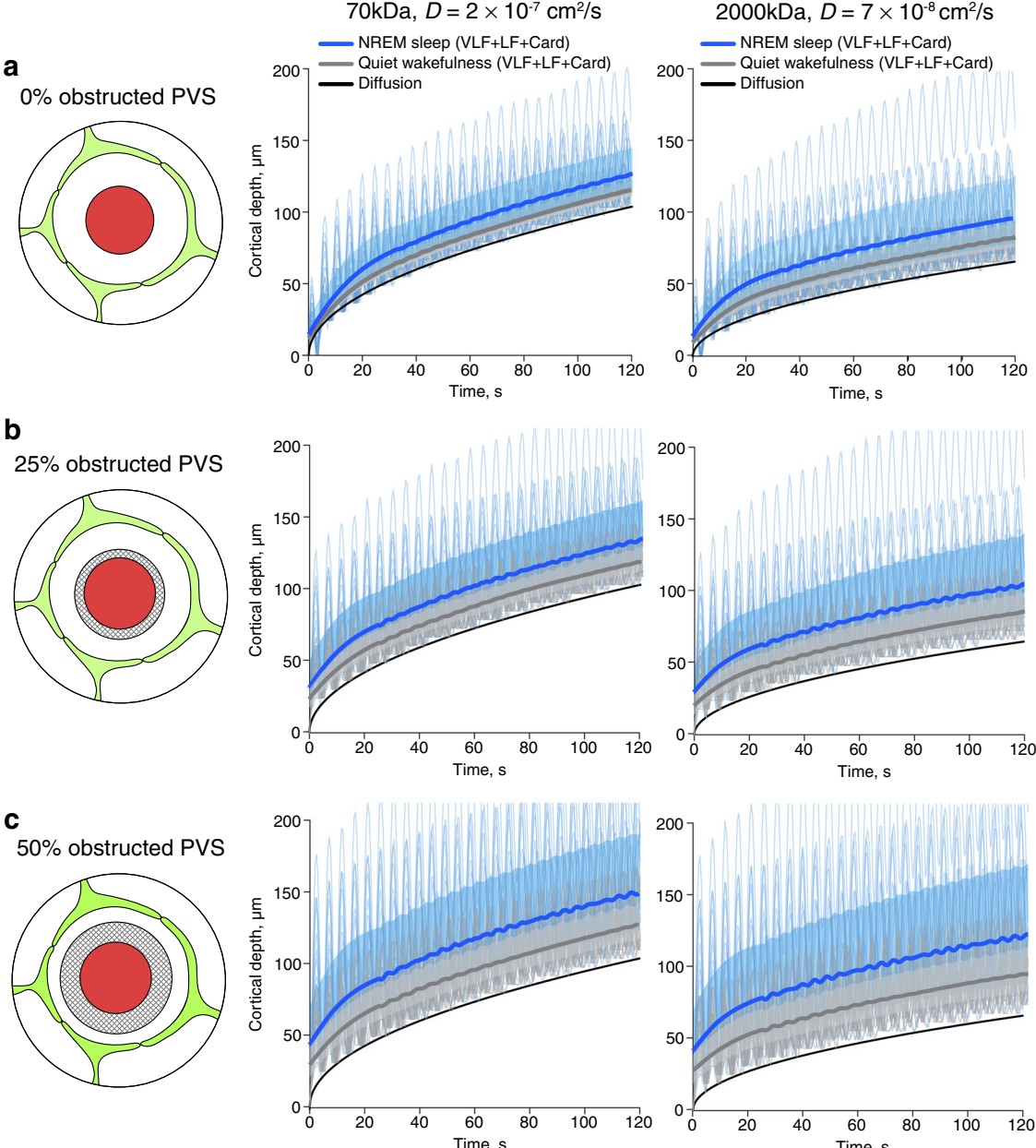

**Fig. 5 | NREM slow vasomotion enhances predicted solute influx. a–c** Influx of 70 and 2000 kDa solutes from the entrance of the penetrating arteriole PVS at the brain surface driven by dispersion during quiet wakefulness, dispersion during NREM sleep or pure diffusion as predicted by biomechanical modeling using three scenarios where we added a fixed volume to the PVS obstructing flow corresponding to 0, 25, and 50% of our measured PVS in quiet wakefulness. The model used the measurements of PVS size for LF and VLF oscillations and assumed a peak CSF velocity of 50 μm/s for the cardiac oscillations. Thin lines represent observations from individual penetrating arterioles ($n = 16$ vessels, 4 mice). Bolded lines with the shading are median values of the time-smoothed thin lines over several oscillations with 10th and 90th percentiles. WBS wake before sleep; NREM non-rapid eye movement sleep; IS intermediate state sleep; REM rapid eye movement sleep; WAS wake after sleep; PVS perivascular space; VLF very low frequency; LF low frequency. Source data are provided as a Source Data file.

PVS. As CSF flow within the ventricles has been shown to be reversed during brain-wide hyperemic patterns in NREM sleep[17], it is possible that the REM sleep-associated vessel dilation and PVS shrinkage would drive CSF out of the brain.

In addition to changes in vascular tone during sleep, we found prominent movement of the astrocytic endfeet that sometimes correlated well with vascular tone, and other times not (Fig. 2c). For instance during the pronounced vasodilation upon REM sleep, the endfoot sleeve also dilated. This could suggest that the size of the free fluid-filled PVS goes toward zero, and potentially there could be direct interactions between vessel wall and endfoot sleeve. The biological observation that the astrocytic endfoot sleeve can dynamically change

its diameter has to the best of our knowledge not been clearly reported before. This again could have profound consequences for fluid movement in the PVS. If ignoring the fluid in the vessel wall, one could be in a situation where the PVS completely exchanges its CSF through a cycle of REM sleep and a subsequent awakening.

The PVS is not an empty space and contains different cell types and extracellular components[11,25,26]. The precise neuroanatomical pathways where CSF and solutes flow, for instance if it is only the fluid filled PVS that allows for fluid and solute movement, or whether there are pathways in between the vascular smooth muscle cells, is still a matter of debate[11]. Moreover, CSF fluorescent tracers, at least under certain experimental protocols, label the entire compartment between

endfoot sleeve and vessel lumen and the resolution limit of optical microscopy does not allow to separately interrogate the different routes. For these reasons we defined PVS as the void between the lumen and endfoot sleeve to encompass all the potential pathways for CSF and solute flow in two-photon image recordings. However, the degree of cellular and non-cellular constituents residing in this space will not only hamper free fluid movement, but also affect the estimated relative change of the fluid filled PVS. Therefore, we also simulated scenarios where we added a fixed volume to the PVS obstructing flow corresponding to 25% and 50% of our measured PVS in quiet wakefulness. In these cases the effect of VLF and LF on CSF flow velocity in PVS (Fig. 3), solute dispersion in PVS (Fig. 4 and Supplementary Fig. 10) and solute influx into PVS (Fig. 5) is considerably larger compared to a situation where we consider no obstruction of flow.

Conversely to REM, during the brief awakenings immediately after REM sleep or the microarousals in NREM and IS sleep, when the vessel constricts and PVS enlarges, one could conjecture that CSF would flow into the PVS of the penetrating arteriole. Such a direct coupling between CSF flow direction and vascular diameter has been shown in a mouse model of ischemia, where vasoconstriction upon spreading depolarization caused a large influx of CSF into the brain[30]. This finding might seem somewhat surprising, given the detrimental effects of wakefulness on glymphatic flow[16]. However, it is important to note that here we describe vascular dynamics upon brief awakenings, also called microarousals, that last from a few up to tens of seconds[21]. While extensive arousals during the night are associated with health hazards[31], microarousals are a normal part of intra-sleep architecture. Such microarousal associated-vasoconstriction is a brief, singular event and apart from exchanging a portion of the CSF volume in the PVS may not contribute to a net flow of CSF into the tissue, as this flow would be dependent on the presumably sleep-wake dependent resistance to flow into the parenchyma.

The model of glymphatic waste clearance consists of three main steps[10], namely influx of CSF along penetrating arterioles, ISF and solute movement through parenchyma and finally efflux of fluid and solutes out of the brain. Here we have only addressed the first step of this process, namely the influx of CSF and solutes into the brain. Increased influx and solute mixing in the PVS during sleep, as we observe, is also likely to enhance parenchymal transport and efflux as PVS flow is important to maintain steep gradients of solutes that are required for efficient solute transport in the parenchyma, yet future studies will have to confirm that. Moreover, the consequences of the difference in vessel dynamics between the arterial side and venous side on solute movement and waste clearance need to be addressed in future studies. Overall, our data adds mechanistic insight into why the glymphatic system is more efficient during sleep. We hypothesize that the entire sleep cycle is required for efficient fluid exchange and solute transport with each part of the sleep cycle playing a different role in the process.

Future studies should address 3 main questions arising from this study. First, whether sleep cycle-dependent vascular dynamics are global throughout the brain and whether they are synchronous along the vascular tree, or propagate in a proximal-to-distal direction or vice versa. This could depend on the type of arteriole PVS[25]. Second, how sleep cycle-dependent vascular dynamics are regulated. For instance, one potential effector for vascular dynamics could be fluctuations in norepinephrine levels across the sleep cycle[32] and its interplay with astrocytic endfeet, that were recently shown to generate ultra-slow arteriole oscillations in awake mice[33]. Third, how vascular dynamics affect intraparenchymal waste clearance.

## Methods
### Animals
Male *Glt1*-eGFP mice[34] were housed on a 12 h light/dark cycle (lights on at 8 AM), 1–4 mice per cage. This mouse strain was originally kindly provided by Professor Jeffrey D. Rothstein (Johns Hopkins University, Baltimore), and has been backcrossed into a C57BL/6J background for 10 generations. Each animal underwent surgery at the age of 8–10 weeks, followed by accommodation to be head-restrained for 1–2 weeks, and two-photon imaging 2–3 times per week for up to 2 months. All imaging experiments were performed on mice that were 10–20 weeks old. Adequate measures were taken to minimize pain and discomfort. The temperature and humidity were controlled, the cages were individually ventilated. Sample sizes were determined based on our previous studies using similar techniques (no power calculations were performed). All procedures were approved by the Norwegian Food Safety Authority (project number: 11983 and 22187).

### Surgical procedures
Mice were anesthetized with isoflurane (3% for induction, 1.5–1.8% for maintenance). Two silver wires (200 μm thickness, non-insulated, GoodFellow) were inserted epidurally into 2 burr holes overlying the right parietal hemisphere for ECoG recordings, and two stainless steel wires (50 μm thickness, insulated except 1 mm tip, GoodFellow) were implanted in the nuchal muscles for EMG recordings. The skull over the left hemisphere was thinned for intrinsic signal imaging, a custom-made titanium head-bar was glued to the skull and the implant sealed with a dental cement cap. After two days, intrinsic optical signal imaging was used to locate the barrel cortex. Individual whiskers were stimulated by a 10 Hz 6 s deflection and representations of individual whiskers in the barrel cortex were detected as increased red light absorption due to increased blood flow to the region. After two days, chronic window implantation was performed by placing a round craniotomy of 2.5 mm diameter centered over the barrel cortex using the location of individual whisker barrels from intrinsic optical imaging as reference. A window made of 2 circular coverslips of 2.5 and 3.5 mm was glued together by ultraviolet curing glue, was then centered in the craniotomy, and fastened by dental cement. Mice with implant complications were excluded from the study.

### Behavioral training
Mice were housed in an enriched environment with a freely spinning wheel in their home cages. One week before imaging, mice were habituated to be head-fixed on a freely spinning wheel under the two-photon microscope. Each mouse was trained head-fixed daily before the imaging for increasing durations ranging from 10 min on the first day to 70 min on the last. Mice that showed signs of stress and did not accommodate to being head-restrained were not included in the study.

### In vivo two-photon imaging
Images and linescans were recorded in ScanImage (Vidrio Technologies) using a custom-built two-photon microscope (Independent NeuroScience Services) equipped with a MaiTai DeepSee ePH DS (Spectra-Physics) laser. A Nikon 16 × 0.8 NA water immersion objective (model CFI75 LWD 16XW) and an excitation wavelength of 920 nm was used for imaging. Linescans were sampled at 250 Hz (149 trials) and 333 Hz (49 trials). Excitation wavelength of 920 nm was used to capture images (512 × 512 pixels) at 30 Hz in the most superficial layer for pial arteries and up to 200 μm below dura for penetrating arterioles and venules of the barrel cortex. Emitted light was routed by a 565 nm longpass dichroic mirror through a 510/80 nm bandpass filter (green channel) or a 630/75 nm bandpass filter (red channel), respectively, and detected with GaAsP amplified photomultiplier tubes from Thorlabs (PMT2101). The vasculature was outlined by 2% 70 kDa Texas Red-labeled dextran (Thermo Fisher Scientific) in saline injected through a tail vein catheter (200 μl at the start of imaging and 50 μl after 5 h to ensure sufficient fluorescence throughout the entire experiment). Pial arteries, arterioles and venules were distinguished based on anatomy and blood flow direction. Head-fixed sleep protocol is described in detail in Bojarskaite et al.[22]. The imaging sessions of sleeping mice

started at 9–10 a.m. (ZT 1–2) and lasted until 3–6 p.m. (ZT 7–10). The mice were allowed to freely move on a disc-shaped wheel/treadmill while awake. Once falling asleep the stage was locked to provide a stable platform for sleep. The position of the disc was adjusted to assist sleep in a head-fixed position. Mice that did not show any signs of sleep within the first 2 h of head-fixation were removed from the microscope. Mice were not sleep-deprived or manipulated in any other way before imaging to induce sleep.

## Electrophysiological and behavioral data acquisition

ECoG and EMG signals were recorded using a DAM50 (WPI) amplifier, denoised by HumBug Noise Eliminators (Quest Scientific) and digitized by a National Instruments input/output device (PCIe-6351). Mouse behavior was recorded by an infrared-sensitive surveillance camera and running wheel motion. Experiments were triggered and synchronized by a TTL signal generated in LabVIEW (LabVIEW 2019, National Instruments) using a National Instruments input/output device (PCIe-6351).

## Sleep-wake state scoring

Wakefulness states were identified using the IR-sensitive surveillance camera video by drawing ROIs over the mouse snout and speed of the running wheel (Supplementary Fig. 1a). The signal in the snout ROIs was quantified by calculating the mean absolute pixel difference between consecutive frames in the respective ROIs. Voluntary locomotion was identified as signals above a threshold in the wheel speed time series. Spontaneous whisking was defined in the snout ROI. Quiet wakefulness was defined as wakefulness with no signal above threshold in snout ROIs and in the wheel speed time series. Sleep states were identified from filtered ECoG (0.5–30 Hz) and EMG signals (100–1000 Hz) based on standard criteria for rodent sleep[20,24] (Supplementary Fig. 1b): NREM sleep was defined as high-amplitude delta (0.5–4 Hz) ECoG activity and low EMG activity; IS was defined as an increase in theta (5–9 Hz) and sigma (9–16 Hz) ECoG activity, and a concomitant decrease in delta ECoG activity; REM sleep was defined as low-amplitude theta ECoG activity with theta/delta ratio >0.5 and low EMG activity. The wakefulness-before-sleep (WBS) episodes were marked as -15 s episodes of behavioral quiescence right before NREM sleep. The wakefulness-after-sleep (WAS) episodes were marked as 10 s episodes starting immediately after awakening from REM sleep when the vessel lumen diameter stabilized. WAS typically contains mouse movement (locomotion, twitching and grooming). Microarousals during NREM and IS sleep were scored as decreases in ECoG total power coupled with activation in EMG signal with a duration of at least 1 s but shorter than 10 s.

## Lumen and endfoot tube diameter and PVS width extraction

All data were managed with a MATLAB-based data management and analysis toolbox Begonia[35]. Line scans were recorded across penetrating arterioles, pial arteries, and veins. The linescans were cropped to center the vessel in the recordings and trials with insufficient signal quality were excluded. To improve signal quality the x-t data were spatiotemporally downsampled by averaging an integer number of samples to most closely reach a sampling frequency of 100 Hz and 20 samples per micrometer. To detect the radius of the lumen and the endfoot tube we created a custom MATLAB tool to manually adjust thresholds throughout the scan with a live preview for each trial to ensure an accurate tracing of both compartments. The manually specified thresholds were linearly interpolated between the chosen threshold-time point pairs. The PVS thickness was calculated by subtracting the lumen diameter from the endfoot tube diameter (Fig. 2b). The endfoot diameter, lumen diameter, and PVS distance on each side of the vessel along with the specific sleep and wake states at each frame were exported to CSV files for further analysis.

## Lumen and endfoot tube diameter and PVS width frequency analyses

High frequency noise was removed using the Savitzky–Golay filter with a time window of 0.1 s and a polynomial fit order of 3. The signal was decomposed into five frequency bands using Butterworth bandpass filters of order 3: continuous (0–0.1 Hz), very low frequency (VLF 0.1–0.3 Hz), low frequency (LF 0.3–1 Hz), respiratory (1–4 Hz) and cardiac (4–15 Hz) (Supplementary Fig. 15a, b). In each frequency band, local minima and maxima were detected in order to identify each individual oscillation (Supplementary Fig. 15c). The signal difference between two consecutive local minima and maxima is referred to as the peak-to-trough (P–T) amplitude. This value then represents the amplitude change in vessel diameter. The time difference between two consecutive local maxima is referred to as the P–P period.

The amplitude of cardiac and respiratory oscillations in lumen of pial arteries and penetrating arterioles are shown in Supplementary Figs. 16 and 17. The period of cardiac lumen oscillations was used for mathematical modeling described below. The oscillation amplitudes for endfoot tube and PVS of penetrating arterioles in respiratory and cardiac frequencies was below the resolution limit, and we could not with certainty rule out effects on rigid movements of the vessels in our scans affecting size estimates from our recordings and therefore not used for further mathematical modeling analyses. For cardiac modeling we have used a range of values to cover a plausible range of oscillatory fluid flow velocities based on the literature.

## Fluid dynamics and solute transport analysis

From the statistical analysis of the P–T amplitude, P–P period and median radius, we performed computer simulation to predict the fluid flow and solute transport in the PVS. The PVS was modeled as the space between two concentric cylinders of length 600 μm, using polar coordinates in 2D where pulsations were induced as radial changes of the inner radius. The CSF was assumed as a Newtonian, incompressible fluid with constant viscosity. The flow was described using Stokes equations and the mass transport was described using the advection diffusion equations:

$$
\begin{aligned}
\mu \nabla^2 \mathbf{u} - \nabla p &= 0, \\
\nabla \cdot \mathbf{u} &= 0, \\
\frac{\partial c}{\partial t} &= (\mathbf{u} \cdot \nabla)c + D\nabla^2 c,
\end{aligned}
\tag{1}
$$

posed in the axisymmetric time-dependent domain representing the model PVS. Here, $u$, $p$, and $c$ are the fluid velocity, pressure and tracer concentration we are predicting. The fluid viscosity was taken as water viscosity at 35 °C, $\mu = 0.693$ mPa/s.

An upper estimate for arteriole PVS thickness from our recordings is $h = 10$ μm (the maximum in our dataset being 8.4 μm), an upper estimate for oscillation frequency is $f = 15$ Hz (maximum in dataset is 14.6 Hz for cardiac frequency) and an upper estimate for fluid velocity is 200 μm/s (our maximal estimate from LF oscillations in the case where 50% of the PVS is obstructed is 177 μm/s). Using those upper estimates we can compute estimates of dimensionless numbers. The maximum Womersley number is then $Wo = h\sqrt{\frac{2\rho\pi f}{\mu}} = 0.1$, and the maximum Reynolds number is $Re = \frac{\rho u h}{\mu} = 3 \times 10^{-3}$, with the density $\rho = 1\,\mathrm{g/cm^3}$ and the dynamic viscosity $\mu = 7 \times 10^{-4}$ Pa. We can therefore neglect in all cases the inertial terms.

The coefficient of diffusion, $D$, depends on the molecular size. For Dextran 70 kDa, the apparent diffusion $D^* = D\lambda^2$ measured in brain neuropil is $0.84 \times 10^{-7}\,\mathrm{cm^2/s}$[36]. Assuming the typical value of a tortuosity $\lambda = 2$ leads to the molecular diffusion $D = 1.7 \times 10^{-7}\,\mathrm{cm^2/s}$. For Dextran 2000 kDa we considered the molecular diffusion coefficient to be $D = 6.8 \times 10^{-7}\,\mathrm{cm^2/s}$[37].

The radius of the internal cylinder $R_v(t)$ is assumed to be uniform along the vessel and dependent on time only. The radius of the

external cylinder $R_{\text{ast}}$ is assumed to be fixed. We assume that both the vessel wall and the astrocyte endfoot tube is impermeable and impose a no flow condition across. The variation of the cross-section area of the PVS $A_{\text{pvs}}(t) = \pi\left(R_{\text{ast}}^2 - R_v^2\right)$ is deduced from the imaging data. It has the form $A_{\text{pvs}}(t) = <A_{\text{pvs}}>\left(1 + \frac{<a_{P-T}>}{2}\cos(\frac{2\pi}{<p_{P-T}>}t)\right) - A_{\text{SMC}}$, with $<A_{\text{pvs}}>$ and $<a_{P-T}>$ being the median of PVS cross section area and median of the P–T amplitude of the cross section area oscillation respectively, and $<p_{P-P}>$ being the median of the lumen P–P period (Supplementary Figs. 17 and 18) for a given vessel, frequency band, and sleep stage. The term $A_{\text{SMC}}$ is an estimate of the cross section area occupied by cellular and non-cellular compartments (most importantly the vessel wall) that do not allow free fluid flow, such as the vessel wall. We assumed these compartments to be deformable but incompressible, thus conserving a fixed volume. The length of the vessel was also assumed to remain constant. The cross section area of the vessel wall was therefore chosen as a constant parameter $A_{\text{SMC}} = \alpha <A_{\text{pvs}}>_{\text{baseline}}$, with $<A_{\text{pvs}}>_{\text{baseline}}$ the median cross section area of the region between the lumen innerside and the astrocyte endfeet measured during baseline, and with $\alpha$ the proportion of this region occupied by cells and non-cellular components found in the PVS. Three values of $\alpha$ were considered: 0%, 25%, and 50%.

For the non-obstructed PVS scenario, the predicted pressure differences along the arteriole PVS are 3 and 5 Pa in quiet wakefulness, and 5 and 7 Pa in NREM state for VLF and LF, respectively (Supplementary Fig. 19a). The pressure gradient 7 Pa/0.6 mm = 11.7 Pa/mm is 13.4 times larger than the estimated pressure gradients of pial PVS (0.8 Pa/mm[38]) and 11.4 times less than the upper estimate of interstitial pressure gradients (133 Pa/mm[39]). Corresponding numbers with 25% obstruction are 21.3 Pa/mm (VLF) and 32 Pa/mm (Supplementary Fig. 19b) and for 50% obstruction 89.0 Pa/mm and 118.3 Pa/mm (Supplementary Fig. 19c).

From the PVS cross section area law $A_{\text{pvs}}(t)$, we then impose the corresponding internal cylinder radius $R_v(t)$ and fluid velocity on the lumen wall $u_y = dR_v/dt$ in the simulation. The fluid domain geometry is updated accordingly at each time step. We impose a reference pressure $p = 0$ at the surface entrance of the PVS and a no flow condition $\mathbf{u} = (0,0)$ at the inner side of the brain.

The equations were discretized using the finite element method in space and an implicit backward Euler scheme in time. The simulations were performed with a maximal time step of $5 \times 10^{-3}$ s and a maximal cell diagonal size of 1 μm.

In the dispersion analysis series, the objective was to estimate the apparent diffusion coefficient by fitting the analytical solution of the 1D pure diffusion equation. We considered the case of the diffusion of an initial Dirac delta distribution of the concentration in the middle of the PVS. The analytical solution has the form

$$c(x,t) = \frac{c_0}{\sqrt{4\pi Dt}}\exp\left(\frac{-(x - L/2)^2}{4Dt}\right) \qquad (2)$$

We chose $c_0 = \sqrt{2\pi\sigma^2}$ and assumed that the initial Dirac delta distribution was set at the time $t = \frac{\sigma^2}{2D}$. The concentration profile used at time $t = 0$ in our simulation has therefore the form

$$c(x,t) = \exp\left(\frac{-(x - L/2)^2}{2\sigma^2}\right) \qquad (3)$$

with $\sigma = 2$ μm being the standard deviation of the Gaussian profile. We then assumed that the concentration profile when the vessel wall velocity is zero remains a gaussian profile and follows the pure diffusion analytical solution. The value of the apparent diffusion $D_{\text{eff}}$ is determined by fitting the analytical solution to the simulation results at $t = 40$ s.

In the present study, we were only able to assess PVS size dynamics for the oscillations in the LF and VLF time scale (due to lack

of imaging resolution). For cardiac oscillations, we, therefore, modeled three scenarios with cardiac oscillations driving CSF fluid peak velocities at 10, 50, and 100 μm/s, and imposed the associated PVS cross section area change.

In the tracer transport analysis series, the concentration was initialized at 0 in the PVS and we imposed a concentration of 1 at the surface entrance of the PVS. We then followed the spread of the tracer with time. We defined the concentration front as the location in the PVS where the concentration had reached 0.1.

## Statistical analyses

Each response variable, for instance the PVS amplitude in the LF frequency band, was analyzed with similar statistical methods. Because of the existence of a global rigid motion of the tissues, the displacements on both sides of a single structure (vessel or endfoot tube) should have a strong positive correlation. If not, this indicates that one edge of the structure was not well detected by our data processing tool. Unrealistic observations were therefore filtered out based on the correlation coefficient (0.8 for lumen and 0.7 for endfoot tube) between the position of both sides of the structure (vessel or astrocytes endfoot tube). Then, the data were aggregated by taking the average of observations belonging to each episode. Here an episode is defined as a continuous time period within a trial where the mouse is identified as behaving according to a single sleep-wake state. A linear mixed effect model was then fitted to the aggregated data set, after log transforming the response variable. For each response variable a sequence of successively simpler models was attempted until reaching numerical convergence and satisfactory looking residual plots. All candidate models had sleep-wake state as a categorical fixed effect, with quiet wakefulness as the baseline state (i.e., the intercept). The models for amplitude and period also included the median lumen radius as a continuous second-order fixed effect, as long as there was sufficient data to estimate these effects. For some response variables, for example for lumen LF amplitude, there was a significant relationship between the response (here amplitude) and median lumen radius. This relationship took different forms, but could often take the form of an inverted U shape, with high expected amplitude for middling median lumen radii, and lower expected amplitude for both small and high lumen radii. All candidate models included random intercepts for each vessel (or penetrating arteriole). For most responses, there was large variation between vessels (as seen for example in the gray lines in Fig. 1d, e.). For response variables where more complex models could be fit (due to sufficient data and depending on residual plots) we also included random effects on the state effect, meaning that each vessel could potentially have somewhat different state-to-state differences. Further, the more complex models also include the possibility of heterogeneous residual variance, in each sleep-wake state and for each mouse. After model fitting, pairwise contrasts between each state estimate were computed. For each fitted model, the p-values belonging to the multiple resulting contrasts were adjusted by Tukey's method for multiple comparisons. All statistical tests were two-sided. Statistical analyses were conducted in R (version 4.0.5). The linear mixed effect models were fitted using the glmmTMB package[40], residual plots were constructed by the DHARMa package (https://CRAN.R-project.org/package=DHARMa), and contrasts computed by the emmeans package (https://CRAN.R-project.org/package=emmeans).

## Reporting summary

Further information on research design is available in the Nature Portfolio Reporting Summary linked to this article.

## Data availability

Source data are provided with this paper. The raw data generated in this study have been deposited in the NIRD Research Data Archive under accession code 2022.00038[41]. A scripts for downloading all files

can be found at https://github.com/GliaLab/PVS-Sleep-Project[42]. The processed data and results from simulations are available at https://doi.org/10.5281/zenodo.7579700. Source data are provided with this paper.

## Code availability

Code for data management and vascular diameter extraction is available here: https://doi.org/10.5281/zenodo.7540534[42]. Code for vascular diameter change analysis, fluid flow simulations, tracer transport simulations and dispersion analysis: https://doi.org/10.5281/zenodo.7579913[43].

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

## Acknowledgements

This work was supported by the Medical Faculty, University of Oslo, the Olav Thon Foundation, the Letten Foundation, the Research Council of Norway (grants #249988, #271555/F20, #302326 #300305, #301013, #303362), the South-Eastern Norway Regional Health Authority (grant #2016070). We acknowledge the support by UNINETT Sigma2 AS for computational science, grant NN9279K and for making data storage available through NIRD, project NS9021K. Associate Professor Gudmund Horn Hermansen is gratefully acknowledged for his input pertaining statistical modeling, Dr. Aree Witoelar is gratefully acknowledged for his help in analyzing microarousals.

## Author contributions

Conceptualization: L.B., A.V., K.A.M., and R.E. Methodology: L.B., A.V., D.M.B., C.C., M.K., K.A.M., and R.E. Formal analysis: D.M.B., A.V., R.E. and C.C. Investigation: L.B. and K.M.G.B. Visualization: L.B., R.E., A.V., and C.C. Data curation: D.M.B., A.V., and C.C. Software: D.M.B., A.V., C.C., and M.K. Funding acquisition: K.A.M. and R.E. Project administration: L.B., D.M.B., A.V., K.A.M., and R.E. Resources: K.A.M., K.H., and R.E. Supervision: K.A.M and R.E. Writing - original draft and revision: L.B., R.E., A.V., C.C., and K.A.M. Writing - review and editing: K.M.G.B., K.H., M.K., C.C., D.M.B., L.B., R.E., A.V., and K.A.M.

## Competing interests

The other authors declare no competing interests.
