## [Peer Review File · Nature Communications]

Sleep cycle-dependent vascular dynamics and the predicted effects on perivascular cerebrospinal fluid flow and solute transportREVIEWER COMMENTS

Reviewer #1 (Remarks to the Author):

This manuscript documents variations with sleep / wake state of the sizes of blood vessels and perivascular spaces (PVSs) in the brains of mice. The study is motivated by interest in flow of cerebrospinal fluid in PVSs, which can remove brain waste and might be used to deliver drugs. Size fluctuations are expressed in terms of frequency bands corresponding to known neurological and physiological activity: low frequency (0.3 to 1 Hz), very low frequency (0.1 to 0.3 Hz), cardiac (4-15 Hz), and respiratory bands (1-4 Hz). The authors find slow, large-amplitude fluctuations during NREM sleep, less fluctuation in other sleep states, and an enlargement of PVSs upon awakening. Since vessel size fluctuations necessarily move the fluid in the vessels, concomitant solute transport is expected, so the manuscript presents numerical simulations of solute transport driven by the measured size fluctuations. The simulations predict enhancements to the effective solute diffusivity by as much as a third, but more often by a few percent.

In vivo measurements of PVS sizes during natural sleep have never before been achieved in any animal, so the data presented will be highly valuable to researchers and clinicians interested in brain fluid flow. Showing significant differences among sleep states (e.g. NREM vs REM) confirms expectations from recent papers showing strong flows driven by functional hyperemia, which is more prominent during NREM. Data is shown in rich and well-constructed figures. The manuscript is written clearly (though it is quite terse!). Thus, I generally support publication, presuming the authors can address a few questions and comments:

Line 34: Multiple publications have also documented efflux along nerve sheaths, in addition to PVSs around veins and arteries.

Line 253: Exclusion of the unsteady term (du/dt) from the momentum equation should be justified.

Line 311: Cardiac frequencies were apparently modeled differently than others. Presumably the lumen diameter was held constant, a no-slip condition was imposed at the lumen wall, and velocity fields were specified according to the known analytic solution for axial flow between concentric cylinders, with the peak velocities listed in the manuscript. In that case, the simulations would have required solving only the advection-diffusion equation, not the mass or momentum equation. Whether my presumptions are correct or not, sufficient detail should be added so that readers don't have to guess.

A few typos:

Line 302: "cross area" should be "cross-sectional area"

Lines 304-305: "The cardiac driven CSF peak velocities ... ARE expected to be much lower"

Reviewer #2 (Remarks to the Author):

The submitted manuscript by Bojarskaite et al., uses state-of-the-art 2-photon imaging in combination EEG/EMG/video recording to show that changes in vascular volumes during natural sleep-wake transitions drives perivascular CSF inflow. This is an important submission because it is the first study to report that changes in the perivascular volume during transition from wake to different sleep stages, including NREM, intermediary and REM sleep. Biomechanical modelling based on the data collected shows that slow and ultraslow vasomotion during NREM sleep is the key driving factor of glymphatic perivascular flow. The study is technically superb, well-written and timely. However, there is no data on CSF flow. Modeling is great, but it does not allow you to make other than prediction. Therefore, the conclusions need to be toned down. Several other points that need to be addressed.

Major critique:

1. The abstract should list that both pial and penetrating arteries are imaged, since the inclusion of penetrating arteries is a clear strength of the study.
2. Ref. 1 is a review questioning the existence of the glymphatic system. Ref. 2 is a general review focused on the structural pathology of the perivascular spaces in small vessel disease. These references choices seem odd based on the study design and experimental findings. An introduction should refer to general reviews on the concept of the glymphatic system and/or to original reports.
3. Refs 5 and 6 are wrongly cited as studies of anesthetized mice.
4. Turner et al., eLife 2020 first reported vascular changes during natural sleep transitions and should be cited already in the result section.
5. It is incorrect to label the perivascular space the black space positioned between the vascular volume and astrocytic endfeet. The black space also contains the vascular smooth muscle cells that - in arteries - in particular changes volume during vascular constriction and dilation. This caveat should be included in the result sections and discussed in the discussion.
6. Would it be possible to model the volume of the smooth muscle cells surrounding the arteries and thereby predict how much error their contribution adds to the perivascular space measurements in the different state of brain activity?
7. Are the authors really suggesting that perivascular CSF in flow is accelerated upon awakening? No existing data support this claim. Only that awakening strongly inhibits inflow. Could the authors discuss additional mechanisms that could antagonize inflow? It is simplistic to think that brain fluid transport is a simple function of the drivers. Additional regulatory mechanisms may exist for a complex organ as the brain.
8. The modelling shown in figure 3 addresses how VLF and LF oscillations enhance solute movement from the SAS into the PVS of a penetrating artery. However, the biological background for this is not argued sufficiently. Arteries in the SAS are also surrounded by PVS that is continuous with the PVS of penetrating arteries. It is not stated if this is also the case in the model or if the SAS is modelled as an open space.
9. The movement of CSF from SAS to PVS is only a minor part of the full CSF movement along the perivascular space and into the brain. More emphasis should be put on if VLF and LF in NREM sleep enhance solute movement compared to wakefulness within the PVS (extended data fig. 4). From extended fig. 4b it does not seem like VLFs makes a significant difference between the states.
10. At page 4 line 109 the authors conclude that "... our modeling data suggest that VLF and LF oscillations during NREM sleep enhance CSF flow and solute transport within the arteriole PVS to levels comparable to enhancement driven by cardiac oscillations." Does this imply that VLF and LF during NREM only contribute to CSF flow if there are no cardiac pulsations or that cardiac pulsations and VLF and LF adds up?
11. The authors use PVS of arteries to model the effect of VLF and LF on flow and argues that this is linked to waste clearance by the glymphatic system. However, "waste" clearance implies the movement of solutes out of the brain and not into it like it is the case in this model. The authors need to argue how solute flow enhancement in arteries can contribute to waste clearance and how it relates to the reported lack of dynamics on the vein side.
12. I Figure 1f is the diameter significant different with a ***P < 0.001 between WBS, NREM and IS. Is this correct? The values are all close to 22 μm .

Reviewer #3 (Remarks to the Author):

In the manuscript by Bojarskaite and colleagues, the authors have developed an innovative method using 2P-IVM to visualize the vascular and astrocyte endfeet diameters in non-anesthetized mice across the sleep-wake cycle. The authors then utilize a portion of the data collected during these imaging sessions to estimate the spread of solutes within the paravascular space using mathematical modeling across the sleep-wake cycle. The authors conclude that slow, large amplitude oscillations during NREM sleep and dramatic changes in PVS diameters during REM sleep and during awakening may facilitate CSF fluid flow, which may help explain previous findings that suggest that fluid clearance is improved during sleep. Overall, the data is very interesting and adds a new component to

the very complex topic of potential CSF/ISF flow into or out of the PVS of the brain. However, the presentation style of the data and the brevity of the text are barriers to effectively communicating the results of the study.

Major comments:

- 1) There is a lot of data presented, while the associated text is quite abbreviated. This makes it difficult for the reader to understand the motivation for each part of the study and how the authors have interpreted the data. It is not clear why the authors have split the data into only 3 main figures, but have included 6 extended data figures and 9 supplementary figures. The authors should give some thought on how to reorganize the figures to make it easier for the reader to understand the key data.
- 2) The text seems quite supportive of the glymphatic model of a CSF directional bulk flow, however, the data does not appear to be so supportive. In the introduction, the authors state that the glymphatic model suggests that CSF flow exits along veins or arteries. However, this is not correct, the model as referenced (Iliff et al, 2012) suggests efflux along veins only and then to lymphatics. The authors should introduce alternative models for CSF/ISF exchange in the introduction, including those of Helen Cserr (who suggested that PVS could serve as efflux routes, but not likely influx routes under normal conditions) or the recent models of Roxane Carare, who suggests that there may be bidirectional flow along the arteries. The authors, rather than only stating that veins showed little diameter changes, should attempt to model solute dispersion in the PVS of the venules, which would lend support to the idea that fluid flow/solute dispersion appears to occur mostly around arteries (driven by vasomotion) rather than veins.
- 3) It is unclear how the authors have come to the conclusion that pulsatile flow due to cardiac rhythm is at the same order of magnitude as the vasomotion effects that have visualized and modeled in the current study. The authors write that the amplitude of PVS diameter changes was below the resolution limit for the recordings and thus the data was not used for further modeling. It is unclear then why the authors are attempting to model the flow generated from cardiac pulsations in this study. Where did the authors determine the CSF fluid peak velocities of 10, 25 and 100 $\mu\text{m/s}$ from? Did the authors take into account that the measured directional CSF flow rates from Mestre et al 2018 (reference 5) are taken from a surface artery and not a penetrating arteriole?
- 4) Pag 2. Line 68. "total width of the PVS was assessed as the difference between vessel lumen diameter and the endfoot tube diameter". The Author should specify the limitations of this analysis and the incorrect concept of the PVS is an empty space. Several cells or structures that may take up some of this space are ignored in the PVS measurement i.e glia limitans, pial cells, smooth muscle cells, and fibroblasts. Thus, the authors should consider that their measurements of this "space" are overestimated.

Minor comments:

- 1) The title is a bit misleading as the conclusions of enhanced perivascular CSF flow and solute transport are based on modeling, whereas the vascular dynamics are actually measured by 2P-IVM.
- 2) Regarding the surgical procedure and in vivo two-photon imaging, please specify the percent of isoflurane used for the surgery, the coordinates of cranial window for two-photon imaging, and the depth of the penetrating arterioles imaged. Are the same cranial window used for optical imaging and two-photon imaging? What is the whisker deflection utilized for?
- 3) Extended data fig 5. What is meant by cardiac speed in this figure?
- 4) Several figures. How was the statistical analysis performed? Representation of the dashed lines representing the excluded data makes the interpretation of the data difficult. For example is Supp Fig 2a, are there plotted values from only 3 pial arteries?

RESPONSE TO REFEREES

Reviewer #1 (Remarks to the Author):

This manuscript documents variations with sleep / wake state of the sizes of blood vessels and perivascular spaces (PVSs) in the brains of mice. The study is motivated by interest in flow of cerebrospinal fluid in PVSs, which can remove brain waste and might be used to deliver drugs. Size fluctuations are expressed in terms of frequency bands corresponding to known neurological and physiological activity: low frequency (0.3 to 1 Hz), very low frequency (0.1 to 0.3 Hz), cardiac (4-15 Hz), and respiratory bands (1-4 Hz). The authors find slow, large-amplitude fluctuations during NREM sleep, less fluctuation in other sleep states, and an enlargement of PVSs upon awakening. Since vessel size fluctuations necessarily move the fluid in the vessels, concomitant solute transport is expected, so the manuscript presents numerical simulations of solute transport driven by the measured size fluctuations. The simulations predict enhancements to the effective solute diffusivity by as much as a third, but more often by a few percent.

In vivo measurements of PVS sizes during natural sleep have never before been achieved in any animal, so the data presented will be highly valuable to researchers and clinicians interested in brain fluid flow. Showing significant differences among sleep states (e.g. NREM vs REM) confirms expectations from recent papers showing strong flows driven by functional hyperemia, which is more prominent during NREM. Data is shown in rich and well-constructed figures. The manuscript is written clearly (though it is quite terse!). Thus, I generally support publication, presuming the authors can address a few questions and comments:

We thank the reviewer for the positive and constructive feedback. We are very pleased to see that the reviewer generally supports the publication and states that ‘the data presented will be highly valuable to researchers and clinicians interested in brain fluid flow’. We have now addressed the questions and comments according to the reviewer’s concerns. Please see specific answers addressed below.

Line 34: Multiple publications have also documented efflux along nerve sheaths, in addition to PVSs around veins and arteries.

We certainly agree that other CSF exit routes are also important. We have now added the following sentences to the introduction regarding efflux pathways other than PVSs around veins and arteries:

Page 2, line 48 of the revised manuscript:

“A model for brain waste clearance – the glymphatic system as proposed in a seminal study ten years ago¹ – states that cerebrospinal fluid (CSF) flows along pial arteries, enters the brain via PVS of penetrating arterioles, then flows through the parenchyma collecting extracellular waste, before it exits in PVS along veins⁵. Other studies also found evidence for clearance along arteries⁶. From there waste may exit the brain along cranial and spinal nerves, via arachnoid granulations (although recently debated)⁷ and meningeal lymphatic vessels^{8,9}, which all drain into the cervical lymphatic vasculature¹⁰.”

Line 253: Exclusion of the unsteady term (du/dt) from the momentum equation should be justified.

An upper estimate for arteriole PVS thickness from our recordings is $h = 10 \mu\text{m}$ (the maximum in our dataset being $8.4 \mu\text{m}$), an upper estimate for oscillation frequency is $f = 15 \text{ Hz}$ (maximum in dataset is 14.6 Hz for cardiac frequency) and an upper estimate for fluid velocity is $200 \mu\text{m/s}$ (our maximal estimate from LF oscillations in the case where 50% of the PVS is obstructed is $177 \mu\text{m/s}$). Using those upper estimates we can compute estimates of dimensionless numbers. The maximum Womersley number is then $Wo = h \sqrt{\frac{2\rho\pi f}{\mu}} = 0.1$, and the maximum Reynolds number is $Re = \frac{\rho u h}{\mu} = 3 \times 10^{-3}$, with the density $\rho = 1 \text{ g/cm}^3$ and the dynamic viscosity $\mu = 7 \times 10^{-4} \text{ Pa}$. Hence, the unsteady term can be safely omitted. Other studies have similar findings, e.g. Daversin-Catty, Cécile, et al. "The mechanisms behind perivascular fluid flow." *Plos one* 15.12 (2020): e0244442, and Kedarasetti, Ravi Teja, Patrick J. Drew, and Francesco Costanzo. "Arterial pulsations drive oscillatory flow of CSF but not directional pumping." *Scientific reports* 10.1 (2020): 1-12. Thomas, John H. "Fluid dynamics of cerebrospinal fluid flow in perivascular

spaces." *Journal of the Royal Society Interface* 16.159 (2019): 20190572. This explanation is added to page 17, line 447.

Line 311: Cardiac frequencies were apparently modeled differently than others. Presumably the lumen diameter was held constant, a no-slip condition was imposed at the lumen wall, and velocity fields were specified according to the known analytic solution for axial flow between concentric cylinders, with the peak velocities listed in the manuscript. In that case, the simulations would have required solving only the advection-diffusion equation, not the mass or momentum equation. Whether my presumptions are correct or not, sufficient detail should be added so that readers don't have to guess.

We were not able to accurately measure the PVS size dynamics in the cardiac range (very close to, or smaller than the resolution limit). Moreover, we were not able to distinguish to what degree rigid movements affected these small amplitudes. Hence, we modeled different scenarios covering a range of cardiac CSF peak velocities (10, 50, 100 $\mu\text{m/s}$), in a plausible range of peak velocities and what has been modeled for pial blood vessels ($\sim 40\mu\text{m/s}$).

In other words, the cardiac frequencies were not modeled differently, but instead of using real measurements from the vessels in our dataset, we used a plausible range of cardiac velocities from the literature.

The manuscript has been expanded throughout, and a more detailed explanation has now been added on page 8, line 199: "Previous modeling studies have predicted that cardiac oscillations lead to a 100% enhancement of the transport of 70 kDa dextran by dispersion and peak oscillatory CSF flow velocity of 100–150 $\mu\text{m/s}$ ^{27–29}. However, these studies deduced the PVS cross-sectional area change from the vessel wall movement only, assuming the astrocyte endfoot tube as rigid, which is not the case in our recordings (Fig. 2, Supplementary Fig. 3). The cardiac driven CSF peak velocities, when astrocyte deformation is taken into account, are expected to be much lower, on the order of 25–50 $\mu\text{m/s}$ for a 250 μm long PVS²⁷. In the present study, we were not able to assess the oscillatory size changes of the PVS of penetrating arterioles in the cardiac frequencies, as the diameter changes were close to the imaging resolution limit, and that we could not rule out an

effect of rigid movements of the vessel on these small measured amplitudes. Therefore, we modeled three different scenarios with cardiac oscillations driving CSF fluid peak velocities at 10, 50 and 100 $\mu\text{m/s}$, and imposed the associated PVS cross section area change (Supplementary Figs. 13 and 14). The dispersion enhancement factor for such cardiac velocities is then of the same order of magnitude as to what we find for the VLF and LF oscillations during NREM sleep (for 0% obstructed PVS: LF/VLF - 6.7% and 29.1% while 50 $\mu\text{m/s}$ cardiac - 5.4% and 13.6%, for 70 kDa and 2000 kDa, respectively) (Fig. 4b and Supplementary Figs. 10 and 13). Overall, our simulations predict that VLF and LF slow oscillations that are of largest amplitude during NREM sleep enhance dispersion of solutes within the arterial PVS to levels comparable to cardiac driven oscillations.“

A few typos:

Line 302: "cross area" should be "cross-sectional area"

Lines 304-305: "The cardiac driven CSF peak velocities ... ARE expected to be much lower"

The typos have been corrected in the revised manuscript.

Reviewer #2 (Remarks to the Author):

The submitted manuscript by Bojarskaite et al., uses state-of-the-art 2-photon imaging in combination EEG/EMG/video recording to show that changes in vascular volumes during natural sleep-wake transitions drives perivascular CSF inflow. This is an important submission because it is the first study to report that changes in the perivascular volume during transition from wake to different sleep stages, including NREM, intermediary and REM sleep. Biomechanical modelling based on the data collected shows that slow and ultraslow vasomotion during NREM sleep is the key driving factor of glymphatic perivascular flow. The study is technically superb, well-written and timely. However, there is no data on CSF flow. Modeling is great, but it does not allow you to make other than prediction. Therefore, the conclusions need to be toned down. Several other points that need to be addressed.

We are pleased to see the reviewer's positive assessment of our work. We agree that modeling can only make predictions of the effects of the vascular dynamics on CSF flow and have modified the conclusions accordingly. We changed the title of the manuscript from "Sleep cycle-dependent vascular dynamics enhance perivascular cerebrospinal fluid flow and solute transport" to "Sleep cycle-dependent vascular dynamics enhance predicted perivascular cerebrospinal fluid flow and solute transport", and made the following changes in the text:

1. Page 2, line 26 (new text underlined) "By biomechanical modeling we demonstrate that these sleep cycle-dependent PVS dynamics likely drive fluid flow and solute transport. These results identify a sleep cycle-dependent mechanism which potentially contributes to enhancement of waste clearance during sleep and can be utilized for CNS drug delivery"
2. Page 6, line 171: "These simulations predict a likely salient role of NREM and IS sleep specific slow vasomotion as a driving force for fluid flow in the PVS in addition to cardiac vessel oscillations."
3. Page 9, line 214: "Overall, our simulations predict that VLF and LF slow oscillations that are of largest amplitude during NREM sleep enhance dispersion of solutes within the arterial PVS to levels comparable to cardiac driven oscillations."
4. Page 10, line 238: "To conclude, our modeling data predict that VLF and LF oscillations increase solute influx in the arteriolar PVS compared to pure diffusion and that the enhancement of solute influx is greatest during NREM sleep when VLF and LF are of largest amplitude."
5. Page 10, line 248: "Using biomechanical modeling we predict that slow, large-amplitude oscillatory vasomotor patterns in NREM sleep are able to generate oscillatory fluid movement on a similar magnitude to heartbeat generated fluid movement in the PVS and enhance solute transport in PVS."

Major critique:

1. The abstract should list that both pial and penetrating arteries are imaged, since the inclusion of penetrating arteries is a clear strength of the study.

We agree that this is a clear strength of the study, and have changed the abstract in the revised manuscript accordingly: page 2, line 21 “Using two-photon imaging of naturally sleeping mice we demonstrate sleep cycle-dependent vascular dynamics of pial arteries and penetrating arterioles: slow, large-amplitude oscillations in NREM sleep, a vasodilation in REM sleep, and a vasoconstriction upon awakening at the end of a sleep cycle. These vascular dynamics are mirrored by changes in the size of the PVS of the penetrating arterioles: slow fluctuations in NREM sleep, reduction in REM sleep and an enlargement upon awakening after REM sleep” .

2. Ref. 1 is a review questioning the existence of the glymphatic system. Ref. 2 is a general review focused on the structural pathology of the perivascular spaces in small vessel disease. These references choices seem odd based on the study design and experimental findings. An introduction should refer to general reviews on the concept of the glymphatic system and/or to original reports.

We have exchanged the references in question to the original paper on the glymphatic system and a recent review on brain fluid flow and glymphatic system, namely:

Rasmussen MK, Mestre H, Nedergaard M. Fluid transport in the brain. *Physiol Rev.* 2022;102(2):1025-1151. doi:10.1152/physrev.00031.2020

Iliff JJ, Wang M, Liao Y, et al. A paravascular pathway facilitates CSF flow through the brain parenchyma and the clearance of interstitial solutes, including amyloid β . *Sci Transl Med.* 2012;4(147):147ra111. doi:10.1126/scitranslmed.3003748

3. Refs 5 and 6 are wrongly cited as studies of anesthetized mice.

Regarding ref. 5, we have now changed the sentence on page 2 line 37 from “This process is thought to be facilitated by mechanical forces created by the vasculature⁴, such as heartbeat driven pial artery pulsations seen in experiments with anesthetized mice⁵ or vasomotion of longer time scales observed in wakefulness⁶.” to “Specifically, heartbeat driven pial artery pulsations have been demonstrated to propel fluorescent microspheres along pial vessels at the brain surface⁵”. However, regarding ref. 6 we are not entirely sure what the reviewer means. Ref. 6 describes the

role of vasomotion in wakefulness, not in anesthesia and we refer to this paper in our manuscript as “vasomotion of longer time scales observed in wakefulness⁶”.

4. Turner et al., eLife 2020 first reported vascular changes during natural sleep transitions and should be cited already in the result section.

We have now added the following sentences to the revised manuscript referring to the work of Turner et al.:

1. Page 4, line 64: “Recently, sleep state coupled changes in brain perfusion and blood flow dynamics of large surface vessels have been demonstrated in mice^{13,14}”
2. Page 6, line 114: “These data clearly show that every part of the sleep cycle entails unique pial artery vascular dynamics and is in line with previous reports on hemodynamic aspects of sleep (Turner et al.)”

5. It is incorrect to label the perivascular space the black space positioned between the vascular volume and astrocytic endfeet. The black space also contains the vascular smooth muscle cells that - in arteries - in particular changes volume during vascular constriction and dilation. This caveat should be included in the result sections and discussed in the discussion.

We have added the following sentence in the results and discussion sections:

Page 6, line 124 of the revised manuscript: “However, defining PVS as the void between lumen and endfoot sleeve will over-estimate the actual PVS volume, as there are cellular and non-cellular constituents within this compartment, such as smooth muscle cells, macrophages, fibroblasts, and extracellular matrix proteins^{25,26}, that likely will hinder fluid flow. For example, the wall of penetrating arterioles consists of smooth muscle cells that would contract upon vasoconstriction. We do not expect the total volume of the smooth muscle cells to change during contraction, and similarly we do not expect the other cellular- and non-cellular components of the PVS to change with vessel diameter changes. For this reason the over-estimation of the PVS volume should be constant across the different sleep states and vascular diameters.”

Page 10, line 271: “The PVS is not an empty space and contains different cell types and extracellular components^{11,25,26}. The precise neuroanatomical pathways where CSF and solutes flow, for instance if it is only the fluid filled PVS that allows for fluid and solute movement, or whether there are pathways in between the vascular smooth muscle cells, for instance is still a matter of debate¹¹. Moreover, CSF fluorescent tracers, at least under certain experimental protocols, label the entire compartment between endfoot sleeve and vessel lumen and the resolution limit of optical microscopy does not allow to separately interrogate the different routes. For these reasons we defined PVS as the void between the lumen and endfoot sleeve to encompass all the potential pathways for CSF and solute flow in two-photon image recordings. However, the degree of cellular and non-cellular constituents residing in this space will not only hamper free fluid movement, but also affect the estimated relative change of the fluid filled PVS. Therefore, we also simulated scenarios where we added a fixed volume to the PVS obstructing flow corresponding to 25% and 50% of our measured PVS in quiet wakefulness. In these cases the effect of VLF and LF on CSF flow velocity in PVS (Fig. 3), solute dispersion in PVS (Fig. 4 and Supplementary Fig. 10) and solute influx into PVS (Fig. 5) is considerably larger compared to a situation where we consider no obstruction of flow”.

6. Would it be possible to model the volume of the smooth muscle cells surrounding the arteries and thereby predict how much error their contribution adds to the perivascular space measurements in the different state of brain activity?

We would like to thank the reviewer for the constructive suggestion of modeling the effect of smooth muscle cells changing their cross sectional diameter with the vascular diameter. Accordingly, we have now run extensive new simulations (60'000 computational hours) presenting 2 new scenarios where we added a fixed volume to the PVS obstructing flow corresponding to 25% and 50% of our measured PVS in quiet wakefulness (Figs. 3–5, Supplementary Figs. 9–11 in the revised manuscript).

Intriguingly, adding a fixed volume of cells and non-cellular constituents hindering fluid flow greatly increases the velocity of oscillatory fluid flow in the PVS (Fig. 3), enhances solute

dispersion in PVS (Fig. 4) and enhances solute influx in the PVS (Fig. 5) compared to no obstruction.

7. Are the authors really suggesting that perivascular CSF in flow is accelerated upon awakening? No existing data support this claim. Only that awakening strongly inhibits inflow. Could the authors discuss additional mechanisms that could antagonize inflow? It is simplistic to think that brain fluid transport is a simple function of the drivers. Additional regulatory mechanisms may exist for a complex organ as the brain.

First, we think it is important to highlight that this increase in PVS volume is something we observe momentarily with the brief awakenings which are part of a natural sleep cycle and occur during the circadian night. Moreover, as we have not directly measured CSF/tracer flux during REM-wakefulness we cannot know what direction the flow will have, as long as flow also would be dependent on the resistance to flow distally. As the reviewer here suggests, increased PVS volume may not necessarily lead to increased flow. For instance, as the increased ECS fraction during sleep may be permissive of increased flow across the endfoot sleeve (and potentially along the vascular tree), this may not be the case when the brain transitions to wakefulness.

We have added the following sentences addressing this question to the discussion:

Pager 11, line 288: “Conversely to REM, during the brief awakenings immediately after REM sleep when the vessel constricts and PVS enlarges, one could conjecture that CSF would flow into the PVS of the penetrating arteriole. Such a direct coupling between CSF flow direction and vascular diameter has been shown in a mouse model of ischemia, where vasoconstriction upon spreading depolarization caused a large influx of CSF into the brain³⁰. This finding might seem somewhat surprising, given the detrimental effects of wakefulness on glymphatic flow¹⁶. However, it is important to note that here we describe vascular dynamics upon brief awakenings at the end of sleep cycles that last from a few up to tens of seconds²¹. While extensive arousals during the night are associated with health hazards³¹, microarousals are a normal part of intra-sleep architecture. Although brief awakenings (also called microarousals) can also occur during NREM sleep, in this paper we have only assessed awakenings at the end of a full sleep cycle after a REM episode. Such awakening associated-vasoconstriction is a brief, singular event typically upon

awakening from REM sleep, and apart from exchanging a portion of the CSF volume in the PVS may not contribute to a net flow of CSF into the tissue, as this flow would be dependent on the presumably sleep-wake dependent resistance to flow into the parenchyma.”

8. The modelling shown in figure 3 addresses how VLF and LF oscillations enhance solute movement from the SAS into the PVS of a penetrating artery. However, the biological background for this is not argued sufficiently. Arteries in the SAS are also surrounded by PVS that is continuous with the PVS of penetrating arteries. It is not stated if this is also the case in the model of if the SAS is modelled as an open space.

We have here modeled a source of solute proximal to a segment of a penetrating arteriole, as an open space without any hindrance to fluid or solute movement. We agree that calling this source for the SAS is a bit simplistic, as it does not take into account the constraints for movement between the pial perivascular compartment, the SAS and the PVS of penetrating arterioles. In any case the reference pressure for the modelling is the pressure at the surface of the brain, and the pressures presented should be interpreted as relative to this. We have now changed the text such that we do not name the source for the modelling as the ‘SAS’, but rather as ‘concentration of 1 at the entrance of the penetrating arteriole PVS’.

We have added the following to the revised manuscript:

Page 9, line 221: “We assumed a fluid filled volume with a solute concentration of 1 at the entrance of the penetrating arteriole PVS, and observed how the oscillatory vessel movements moved the solute concentration front (defined as the point with a concentration of 0.1) as it traversed the depth of the PVS.”

9. The movement of CSF from SAS to PVS is only a minor part of the full CSF movement along the perivascular space and into the brain. More emphasis should be put on if VLF and LF in NREM sleep enhance solute movement compared to wakefulness within the PVS (extended data fig. 4). From extended fig. 4b it does not seem like VLFs makes a significant difference between the states.

We agree that from the extended fig. 4b (Supplementary Fig. 10a in the revised manuscript), which is the same data as for Fig. 3d (Fig. 4b in the revised manuscript), it does not seem that VLF makes a significant difference in enhancing dispersion for 70kDa solute between the states. We have now run simulations where we added a fixed volume to the PVS obstructing flow corresponding to 25% and 50% of our measured PVS in quiet wakefulness simulating potential effects of cellular and non-cellular constituents in the PVS that would hinder the flow of fluid and solutes, which is likely a scenario closer to the true physiology as also discussed in question 6 on page 8. With these new simulations we predict that with 25% obstructed PVS VLF leads to 10.8% and 17.3% enhancement in NREM compared to only 5.5% and 9.4% in wakefulness for 70kDa and 2000kDa solute dispersion, respectively (Fig. 4c and Supplementary Fig. 10b); with 50% obstructed PVS VLF leads to 17.4% enhancement and 25% enhancement in NREM compared to only 8.4% and 12.6% in wakefulness for 70kDa and 2000kDa solute dispersion, respectively (Fig. 4d and Supplementary Fig. 10c).

10. At page 4 line 109 the authors conclude that “... our modeling data suggest that VLF and LF oscillations during NREM sleep enhance CSF flow and solute transport within the arteriole PVS to levels comparable to enhancement driven by cardiac oscillations.” Does this imply that VLF and LF during NREM only contribute to CSF flow if there are no cardiac pulsations or that cardiac pulsations and VLF and LF adds up?

The effect of VLF and LF during NREM sleep and the effect of cardiac pulsations on fluid flow and solute transport in arterial PVS add up and are of a comparable magnitude.

11. The authors use PVS of arteries to model the effect of VLF and LF on flow and argues that this is linked to waste clearance by the glymphatic system. However, “waste” clearance implies the movement of solutes out of the brain and not into it like it is the case in this model. The authors need to argue how solute flow enhancement in arteries can contribute to waste clearance and how it relates to the reported lack of dynamics on the vein side.

We agree that discussing waste clearance when we have only modeled influx needs more justification in the text. We chose to model influx as this is representative of imaging studies of

glymphatic flow in the PVS of arteries and arterioles typically using a fluorescent tracer to show influx, with the premise that this fluid flow drives waste clearance through glymphatic flow.

We have added the following sentences to the discussion, page 12, line 304: “Glymphatic waste clearance consists of three main steps, namely influx of CSF along penetrating arterioles, ISF and solute movement through parenchyma and finally efflux of fluid and solutes out of the brain¹⁰. Here with our modeling attempts we have only addressed the first step of this process, namely the influx of CSF and solutes into the brain. Increased influx and solute mixing in the PVS during sleep, as we observe, is also likely to enhance parenchymal transport and efflux as PVS flow is important to maintain steep gradients of solutes that are required for effective solute transport in the parenchyma, yet future studies will have to confirm that. Moreover, the consequences of the difference in vessel dynamics between the arterial side and venous side on solute movement and waste clearance need to be addressed in future studies.”

12. I Figure 1f is the diameter significant different with a ***P < 0.001 between WBS, NREM and IS. Is this correct? The values are all close to 22 μm .

Figure 1f shows the vessel diameters and how they change from different sleep states. However, the statistics is performed looking at the change in diameter for a given vessel, not the absolute value (which is what is represented by the mean trace in black).

The analysis in Figure 1f concerns the *difference in diameter between sleep states within each vessel*. And since the vessels sampled in this study are of a range of different diameters, the scale of the figure renders the differences within each vessel virtually invisible (at least between Baseline, NREM and IS states). If each vessel had been given its own scale (or figure) the differences would have been more visible, but still modest: the estimated model indicates that, on average, the vessels have a 4% larger diameter in NREM than in baseline, and a 7% larger diameter in IS than in baseline (compared to a 28% increase between REM and baseline). The effect size is admittedly not very large for these specific examples, but significant differences are found because the measurements within repeated episodes in the same state and vessel were relatively consistent.

Reviewer #3 (Remarks to the Author):

In the manuscript by Bojarskaite and colleagues, the authors have developed an innovative method using 2P-IVM to visualize the vascular and astrocyte endfeet diameters in non-anesthetized mice across the sleep-wake cycle. The authors then utilize a portion of the data collected during these imaging sessions to estimate the spread of solutes within the paravascular space using mathematical modeling across the sleep-wake cycle. The authors conclude that slow, large amplitude oscillations during NREM sleep and dramatic changes in PVS diameters during REM sleep and during awakening may facilitate CSF fluid flow, which may help explain previous findings that suggest that fluid clearance is improved during sleep. Overall, the data is very interesting and adds a new component to the very complex topic of potential CSF/ISF flow into or out of the PVS of the brain. However, the presentation style of the data and the brevity of the text are barriers to effectively communicating the results of the study.

We are pleased to see the reviewer's positive assessment of our work. We have now significantly expanded the manuscript, re-arranged the figures to improve data presentation and we hope the reviewer will find that the new, considerably longer manuscript is more informative and communicates our results more effectively.

Major comments:

1) There is a lot of data presented, while the associated text is quite abbreviated. This makes it difficult for the reader to understand the motivation for each part of the study and how the authors have interpreted the data. It is not clear why the authors have split the data into only 3 main figures, but have included 6 extended data figures and 9 supplementary figures. The authors should give some thought on how to reorganize the figures to make it easier for the reader to understand the key data.

We have now extensively expanded the manuscript, including explanations for the motivation for each part of the study and more detailed discussion of data interpretation (revised manuscript is 3797 words longer than the original manuscript that was formatted in a brief communications

format), and has 2 additional main figures. We have re-organized the figures to make it easier for the reader to understand key data, and now have 5 main figures, 19 supplementary figures and no extended data.

2) The text seems quite supportive of the glymphatic model of a CSF directional bulk flow, however, the data does not appear to be so supportive. In the introduction, the authors state that the glymphatic model suggests that CSF flow exits along veins or arteries. However, this is not correct, the model as referenced (Iliff et al, 2012) suggests efflux along veins only and then to lymphatics. The authors should introduce alternative models for CSF/ISF exchange in the introduction, including those of Helen Cserr (who suggested that PVS could serve as efflux routes, but not likely influx routes under normal conditions) or the recent models of Roxane Carare, who suggests that there may be bidirectional flow along the arteries. The authors, rather than only stating that veins showed little diameter changes, should attempt to model solute dispersion in the PVS of the venules, which would lend support to the idea that fluid flow/solute dispersion appears to occur mostly around arteries (driven by vasomotion) rather than veins.

We agree with the reviewer that several CSF/ISF exchange models should be introduced. We have now expanded the introduction to include works of Helen Cserr and Roxane Carare, and corrected the description of the canonical glymphatic model in regards to exit pathways. We have made the following edits in the revised manuscript (new text underlined, removed text struck through):

Page 3, line 36: “Already in the 1970s pioneering work of Helen Cserr and colleagues identified the perivascular compartments of the brain as pathways enabling bulk flow and efficient transport of solutes out of the brain⁴. A model for brain waste clearance – the glymphatic system as proposed in a seminal study ten years ago¹ – states that cerebrospinal fluid (CSF) flows along pial arteries, enters the brain via PVS of penetrating arterioles, then flows through the parenchyma collecting extracellular waste, before it exits in PVS along veins⁵. Other studies also found evidence for clearance along arteries⁶. From there waste may exit the brain along cranial and spinal nerves, via arachnoid granulations (although recently debated)⁷ and meningeal lymphatic vessels^{8,9}, which all drain into the cervical lymphatic vasculature¹⁰. In addition, there might exist a bidirectional CSF and solute flow along the arteries⁶”

We have now also modeled solute dispersion in the PVS of the venules:

Page 8, line 195: “The enhancement factor for tracer dispersion in vein PVS was insignificant compared to arteriole PVS and reached maximum values of 5.3% and 1.6% for 2000 kDa and 70 kDa tracers, respectively (Supplementary Fig. 12).”

Page 12, line 310: “Moreover, the consequences of the difference in vessel dynamics between the arterial side and venous side on solute movement and waste clearance need to be addressed in future studies.”

3) It is unclear how the authors have come to the conclusion that pulsatile flow due to cardiac rhythm is at the same order of magnitude as the vasomotion effects that have visualized and modeled in the current study. The authors write that the amplitude of PVS diameter changes was below the resolution limit for the recordings and thus the data was not used for further modeling. It is unclear then why the authors are attempting to model the flow generated from cardiac pulsations in this study. Where did the authors determine the CSF fluid peak velocities of 10, 25 and 100 $\mu\text{m/s}$ from? Did the authors take into account that the measured directional CSF flow rates from Mestre et al 2018 (reference 5) are taken from a surface artery and not a penetrating arteriole?

We have now added a detailed explanation for our conclusion that pulsatile flow due to cardiac rhythm likely is of the same order of magnitude as the slow vasomotion effects observed in our study: Revised manuscript, Results section page 8, line 199.

“Previous modeling studies have predicted that cardiac oscillations lead to a 100% enhancement of the transport of 70 kDa dextran by dispersion and peak oscillatory CSF flow velocity of 100–150 $\mu\text{m/s}$ ^{27–29}. However, these studies deduced the PVS cross-sectional area change from the vessel wall movement only, assuming the astrocyte endfoot tube as rigid, which is not the case in our recordings (Fig. 2, Supplementary Fig. 3). The cardiac driven CSF peak velocities, when astrocyte deformation is taken into account, are expected to be much lower, on the order of 25–50 $\mu\text{m/s}$ for a 250 μm long PVS²⁷. In the present study, we were not able to assess the oscillatory size

changes of the PVS of penetrating arterioles in the cardiac frequencies, as the diameter changes were close to the imaging resolution limit, and that we could not rule out an effect of rigid movements of the vessel on these small measured amplitudes. Therefore, we modeled three different scenarios with cardiac oscillations driving CSF fluid peak velocities at 10, 50 and 100 $\mu\text{m/s}$, and imposed the associated PVS cross section area change (Supplementary Figs. 13 and 14). The dispersion enhancement factor for such cardiac velocities is then of the same order of magnitude as to what we find for the VLF and LF oscillations during NREM sleep (for 0% obstructed PVS: LF/VLF - 6.7% and 29.1% while 50 $\mu\text{m/s}$ cardiac - 5.4% and 13.6%, for 70 kDa and 2000 kDa, respectively) (Fig. 4b and Supplementary Figs. 10 and 13). Overall, our simulations predict that VLF and LF slow oscillations that are of largest amplitude during NREM sleep enhance dispersion of solutes within the arterial PVS to levels comparable to cardiac driven oscillations.“

4) Pag 2. Line 68. “total width of the PVS was assessed as the difference between vessel lumen diameter and the endfoot tube diameter”. The Author should specify the limitations of this analysis and the incorrect concept of the PVS is an empty space. Several cells or structures that may take up some of this space are ignored in the PVS measurement i.e glia limitans, pial cells, smooth muscle cells, and fibroblasts. Thus, the authors should consider that their measurements of this “space” are overestimated.

We have added the following sentence in the results and discussion sections:

Page 6, line 124 of the revised manuscript: “However, defining PVS as the void between lumen and endfoot sleeve will over-estimate the actual PVS volume, as there are cellular and non-cellular constituents within this compartment, such as smooth muscle cells, macrophages, fibroblasts, and extracellular matrix proteins^{25,26}, that likely will hinder fluid flow. For example, the wall of penetrating arterioles consists of smooth muscle cells that would contract upon vasoconstriction. We do not expect the total volume of the smooth muscle cells to change during contraction, and similarly we do not expect the other cellular- and non-cellular components of the PVS to change with vessel diameter changes. For this reason the over-estimation of the PVS volume should be constant across the different sleep states and vascular diameters.”

Page 11, line 271: “The PVS is not an empty space and contains different cell types and extracellular components^{11,25,26}. The precise neuroanatomical pathways where CSF and solutes flow, for instance if it is only the fluid filled PVS that allows for fluid and solute movement, or whether there are pathways in between the vascular smooth muscle cells, for instance is still a matter of debate¹¹. Moreover, CSF fluorescent tracers, at least under certain experimental protocols, label the entire compartment between endfoot sleeve and vessel lumen and the resolution limit of optical microscopy does not allow to separately interrogate the different routes. For these reasons we defined PVS as the void between the lumen and endfoot sleeve to encompass all the potential pathways for CSF and solute flow in two-photon image recordings. However, the degree of cellular and non-cellular constituents residing in this space will not only hamper free fluid movement, but also affect the estimated relative change of the fluid filled PVS. Therefore, we also simulated scenarios where we added a fixed volume to the PVS obstructing flow corresponding to 25% and 50% of our measured PVS in quiet wakefulness. In these cases the effect of VLF and LF on CSF flow velocity in PVS (Fig. 3), solute dispersion in PVS (Fig. 4 and Supplementary Fig. 10) and solute influx into PVS (Fig. 5) is considerably larger compared to a situation where we consider no obstruction of flow”.

We have also run extensive new simulations (~60'000 computing hours) presenting 2 new scenarios compared to the original data where we added a fixed volume to the PVS obstructing flow corresponding to 25% and 50% of our measured PVS in quiet wakefulness (Figs. 3–5, Supplementary Figs. 9–11 in the revised manuscript). Intriguingly, adding a fixed volume of cells and non-cellular constituents hindering fluid flow greatly increases the velocity of oscillatory fluid flow in the PVS (Fig. 3), enhances solute dispersion in PVS (Fig. 4) and enhances solute influx in the PVS (Fig. 5) compared to no obstruction.

Minor comments:

1) The title is a bit misleading as the conclusions of enhanced perivascular CSF flow and solute transport are based on modeling, whereas the vascular dynamics are actually measured by 2P-IVM.

We have now changed the title from “Sleep cycle-dependent vascular dynamics enhance perivascular cerebrospinal fluid flow and solute transport” to “Sleep cycle-dependent vascular dynamics enhance predicted perivascular cerebrospinal fluid flow and solute transport”

2) Regarding the surgical procedure and in vivo two-photon imaging, please specify the percent of isoflurane used for the surgery, the coordinates of cranial window for two-photon imaging, and the depth of the penetrating arterioles imaged. Are the same cranial window used for optical imaging and two-photon imaging? What is the whisker deflection utilized for?

We use 3% isoflurane for induction of anesthesia (2–3 min) and 1.5-1.8% isoflurane for maintenance of anesthesia during the surgery. We have added this information to the Methods section “Surgical procedures” on page 13, line 334 in the revised manuscript (new text underlined): “Mice were anesthetized with isoflurane (3% for induction, 1.5–1.8% for maintenance)”.

We used intrinsic optical imaging to precisely locate the barrel cortex over which we positioned the chronic cranial window. To do this, one whisker was stimulated to increase blood flow to the corresponding barrel region in the somatosensory cortex, which was detected as increased absorption of red light. This method is more reliable to precisely locate the barrel cortex than using anatomical coordinates as for example bregma might appear different from mouse to mouse and this would introduce error in the coordinate measurement. No cranial window was made prior to intrinsic imaging, instead, the skull was thinned. After having identified the barrels, we drilled out the chronic cranial window used for two-photon imaging. We have added detail and clarification to Methods section “Surgical procedures”, page 13, line 333 in the revised manuscript:

We performed the imaging of penetrating arterioles from the surface of the brain up to the depth of 200 μm . We have now adjusted the text in Methods section “In vivo two-photon imaging” (page 14, line 361) to include this information (new text underlined, removed old text striked through): “Excitation wavelength of 920 nm was used to capture images (512 x 512 pixels) at 30 Hz in the

most superficial layer for pial arteries and up to 200 μm below dura for penetrating arterioles and venules of the barrel cortex.”

3) Extended data fig 5. What is meant by cardiac speed in this figure?

Cardiac speed in Extended data fig. 5, which is now Supplementary Figure 13 in the revised manuscript refers to CSF peak velocity generated by cardiac oscillations. This is now more clearly explained in the revised manuscript:

Page 8 line 199: “Previous modeling studies have predicted that cardiac oscillations lead to a 100% enhancement of the transport of 70 kDa dextran by dispersion and peak oscillatory CSF flow velocity of 100–150 $\mu\text{m}/\text{s}$ ^{27–29}. However, these studies deduced the PVS cross-sectional area change from the vessel wall movement only, assuming the astrocyte endfoot tube as rigid, which is not the case in our recordings (Fig. 2, Supplementary Fig. 3). The cardiac driven CSF peak velocities, when astrocyte deformation is taken into account, are expected to be much lower, on the order of 25–50 $\mu\text{m}/\text{s}$ for a 250 μm long PVS²⁷. In the present study, we were not able to assess the oscillatory size changes of the PVS of penetrating arterioles in the cardiac frequencies, as the diameter changes were close to the imaging resolution limit, and that we could not rule out an effect of rigid movements of the vessel on these small measured amplitudes. Therefore, we modeled three different scenarios with cardiac oscillations driving CSF fluid peak velocities at 10, 50 and 100 $\mu\text{m}/\text{s}$, and imposed the associated PVS cross section area change (Supplementary Figs. 13 and 14).”

4) Several figures. How was the statistical analysis performed? Representation of the dashed lines representing the excluded data makes the interpretation of the data difficult. For example is Supp Fig 2a, are there plotted values from only 3 pial arteries?

For all of the figures with statistical comparisons the statistical model used was a linear mixed effect model, as explained, and now expanded in the Methods, Statistical analyses. Tables describing the test statistics and degrees of freedom for all comparisons are added at the end of the supplementary information file. The dashed lines do not represent excluded data, but simply the

fact that each vessel is not necessarily observed in all sleep or wake states: Every separate line, full or dashed, represents an individual blood vessel. For example, in Supplementary Fig. 2a, there are 19 individual lines, meaning 19 individual pial arteries. Dashed line is used when there is a missing state between other 2 joining states, for example, as in Supplementary Fig. 2a, dashed line is used only when data value for “whisking” is missing, but we have values for “quiet wakefulness” and “locomotion”. In contrast, when data is missing for “locomotion”, a full line is used, as you can see two full short lines connecting only “quiet wakefulness” and “whisking” in Supplementary Fig. 2a. In figures showing data from sleep: for example Fig. 2d lumen, there are two dashed lines, one connecting “WBS” and “IS”, that become a full line connecting IS to REM and REM to WAS. This means that we do not have data only for “NREM” sleep. The second line is dashed between NREM and REM, this means that we lack data only for IS, as lines connecting WBS and NREM, and REM to WAS are full lines.

We have throughout added more clarification to the figure legends to aid in understanding the dashed lines in the plots, for example, legend for Fig. 1, page 23, line 596 in the revised manuscript. “Every gray line (dashed or full) represents an individual vessel, dashed lines are used when an observation is missing in a certain state for a given vessel (for example dashed line between WBS and IS means that we do not have an observation for that particular vessel in NREM state), bold lines and shaded area are the estimates and 95% CI from linear mixed effects models.”

REVIEWER COMMENTS

Reviewer #1 (Remarks to the Author):

The authors addressed my concerns. Additionally, the manuscript is notably stronger after being significantly expanded. I support publication.

Reviewer #2 (Remarks to the Author):

The manuscript by Bojarskaite et al uses beautiful 2-photon imaging to show state dependent changes in the vascular network and modeling the predict its effect on periarterial CSF flow. The authors have responded well to the critique. Just a few points remain:

1) Titled of first result section: "Two-photon imaging of PVS dynamics in natural sleep" → "Two-photon imaging of vascular dynamics in natural sleep". Also, why is the GLT1 signal not displayed in Fig. 1?

2) The use of 'predicted' in the title is somehow misleading. It feels like forecasting or like if there were in vivo measurements of CSF flow that supports the calculations. The bottom line is that fluid flow should be measured directly and cannot be predicted based on changes in compartment size if the underlying pressure is unknown. The title could be change to: "Sleep cycle-dependent vascular dynamics and the predicted effects on perivascular cerebrospinal fluid flow and solute transport ". Also, the authors must tone the claims based on modeling down. Their modeling data does not justify the statement: "These results identify a sleep cycle-dependent mechanism which potentially contributes to enhancement of waste clearance during sleep and can be utilized for CNS drug delivery".

3) Line 293: "However, it is important to note that here we describe vascular dynamics upon brief awakenings at the end of sleep cycles that last from a few up to tens of seconds²¹." These shortlasting episodes are normally described as microarousal and not as 'brief awakening'. Also, why are the microarousal only studied when they happened after REM and not NREM?

4) I find Fig. 3 a bit confusing. First, I would suggest changing the orientation of the bottom illustration so that the flow goes down as seen in glymphatic imaging studies (or does the flow in the model flows in the opposite direction? Or does flow directionality even means anything in the model?). Second, I do not understand what the shape and size of the black circle represents. Since the number represents the peak velocity, is the size of the black circle related to the velocity? Or does it describe the size of the PVS?

5) Does the flow from LF and VLF in the model move in any specific direction? From Fig. 3a and Fig. 5 it looks like it follows a direction and 'CSF goes deeper'. Fig. 4a, however, shows that the solute spreads both up and down. Could the authors repeat the solute influx simulations but imposing the concentration of 1 in the middle of the PVS rather than at the surface entrance, and then compared the concentration front up movement up and down the PVSs?

6) Line 125: "there are cellular and non-cellular constituents within this compartment, such as smooth muscle cells, macrophages, fibroblasts, and extracellular matrix proteins^{25,26}, that likely hinder fluid flow." This is logic since smaller space means higher resistance to flow. However, the simulations with the 25% and 50% PVS obstruction showed higher peak velocities that the 0%. How does the author explain this apparent contradiction?

Reviewer #3 (Remarks to the Author):

Thank you to the authors for answering my questions and addressing my concerns. The revised manuscript is much improved and the title is more reflective of the data that is presented in the study. I would just request a few minor changes to the text:

Line 50: The previous paragraph has introduced the different models for flow along the PVS, thus one should rephrase "glymphatic" to "Flow along the PVS" at the beginning of the next paragraph.

Line 256: Clarify that the CSF flow within the ventricles was shown to be reversed in the referenced study.

Line 304: Please preface this sentence with "The model of" glymphatic waste clearance.

REVIEWER COMMENTS

Reviewer #1 (Remarks to the Author):

The authors addressed my concerns. Additionally, the manuscript is notably stronger after being significantly expanded. I support publication.

We thank the reviewer for positive evaluation of our revised manuscript.

Reviewer #2 (Remarks to the Author):

The manuscript by Bojarskaite et al uses beautiful 2-photon imaging to show state dependent changes in the vascular network and modeling the predict its effect on periarterial CSF flow. The authors have responded well to the critique. Just a few points remain:

We thank the reviewer for such a positive evaluation of our manuscript, and appreciate that our response to the critique was well accepted. We addressed the few last points below.

1) Titled of first result section: “Two-photon imaging of PVS dynamics in natural sleep” → “Two-photon imaging of vascular dynamics in natural sleep”. Also, why is the GLT1 signal not displayed in Fig. 1?

We agree with the reviewer that the suggested subtitle is more appropriate, and it has been changed in the revised manuscript.

The GLT1 signal is not displayed in Fig.1 because Fig.1 includes only pial arteries which are not surrounded by astrocytic endfeet, hence there is no GLT1 labeling.

2) The use of ‘predicted’ in the title is somehow misleading. It feels like forecasting or like if there were in vivo measurements of CSF flow that supports the calculations. The bottom line is that fluid flow should be measured directly and cannot be predicted based on changes in compartment size if the underlying pressure is unknown. The title could be change to: “Sleep cycle-dependent vascular dynamics and the predicted effects on perivascular cerebrospinal fluid flow and solute transport“. Also, the authors must tone the claims based on modeling down. Their modeling data does not justify the statement: “These results identify a sleep cycle-dependent mechanism which potentially contributes to enhancement of waste clearance during sleep and can be utilized for CNS drug delivery”.

We agree with the reviewer that the suggested title maybe is more appropriate, and have changed the title accordingly.

The statement *“These results identify a sleep cycle-dependent mechanism which potentially contributes to enhancement of waste clearance during sleep and can be utilized for CNS drug delivery”* has been removed from the manuscript.

Moreover, we have also added a sentence to the discussion (page 10, line 254): *“It is important to stress that these observations from mathematical models have to be confirmed with biological experiments, such as for instance labeling CSF and PVS with fluorescent tracers.”*

3) Line 293: *“However, it is important to note that here we describe vascular dynamics upon brief awakenings at the end of sleep cycles that last from a few up to tens of seconds²¹.”* These shortlasting episodes are normally described as microarousal and not as ‘brief awakening’. Also, why are the microarousal only studied when they happened after REM and not NREM?

We agree that also microarousals within NREM/IS sleep would be interesting to report in addition to the ones ending a sleep cycle. Accordingly, we have now analyzed the relationship between vessel lumen and PVS size during microarousals during NREM and IS sleep as well. Results are included in Supplementary Fig. 3c and described in the results and discussion in the revised manuscript:

Page 2, line 21: *“Using two-photon imaging of naturally sleeping mice we demonstrate sleep cycle-dependent vascular dynamics of pial arteries and penetrating arterioles: slow, large-amplitude oscillations in NREM sleep, a vasodilation in REM sleep, and a vasoconstriction upon awakening at the end of a sleep cycle and microarousals in NREM and IS sleep. These vascular dynamics are mirrored by changes in the size of the PVS of the penetrating arterioles: slow fluctuations in NREM sleep, reduction in REM sleep and an enlargement upon awakening after REM sleep and during microarousals in NREM and IS sleep.”*

Page 4, line 77: *“During the brief arousal at the end of a sleep cycle after REM sleep and microarousals in NREM and IS sleep, arteries and arterioles constrict and PVS enlarges.”*

Page 6, line 145: *“In contrast, upon the brief awakening at the end of a sleep cycle after a REM episode and microarousals in NREM and IS sleep, arteriole lumen and the endfoot tube constricted to reach a similar size as in quiet wakefulness, while the PVS enlarged (Fig. 2e, Supplementary Figs. 3b,c, 5a and Supplementary Tables 3–6)”*

Page 10, line 256: *“The next steps will be to understand how REM sleep specific reduction in the size of PVS and the subsequent enlargement of PVS upon awakening at the end of the*

sleep cycle or during the microarousals in NREM and IS sleep affect fluid flow and solute transport in PVS.”

Page 11, line 291: *“Conversely to REM, during the brief awakenings immediately after REM sleep or the microarousals in NREM and IS sleep, when the vessel constricts and PVS enlarges, one could conjecture that CSF would flow into the PVS of the penetrating arteriole.”*

Page 11, line 297: *“However, it is important to note that here we describe vascular dynamics upon brief awakenings, also called microarousals, that last from a few up to tens of seconds²¹. While extensive arousals during the night are associated with health hazards³¹, microarousals are a normal part of intra-sleep architecture. Such microarousal associated-vasoconstriction is a brief, singular event and apart from exchanging a portion of the CSF volume in the PVS may not contribute to a net flow of CSF into the tissue, as this flow would be dependent on the presumably sleep-wake dependent resistance to flow into the parenchyma.”*

4) I find Fig. 3 a bit confusing. First, I would suggest changing the orientation of the bottom illustration so that the flow goes down as seen in glymphatic imaging studies (or does the flow in the model flows in the opposite direction? Or does flow directionality even means anything in the model?).

The bottom illustration has been changed according to the reviewer comment so that the flow goes down. See also answer for question nr. 5 below.

Second, I do not understand what the shape and size of the black circle represents. Since the number represents the peak velocity, is the size of the black circle related to the velocity? Or does it describe the size of the PVS?

The size of the black circle represents the median of the parameter measured (e.g. Fig. 3 - median velocity, Fig. 4 - median enhancement factor), whereas the shading represents the distribution of all modeled vessels. The median is also given by the numbers in the lower right corner in each box; this information is also now added in the figure legends for figures 3 and 4.

5) Does the flow from LF and VLF in the model move in any specific direction? From Fig. 3a and Fig. 5 it looks like it follows a direction and ‘CSF goes deeper’. Fig. 4a, however, shows that the solute spreads both up and down. Could the authors repeat the solute influx simulations but imposing the concentration of 1 in the middle of the PVS rather than at the

surface entrance, and then compared the concentration front up movement up and down the PVSs?

Fig. 3a was an example illustration and now we have changed the arrows to point downwards. In Fig. 5 we specifically modeled solute movement from the surface of the brain to see whether LF and VLF would contribute to solute influx as compared to pure diffusion.

The flow from LF and VLF in the model does not have a specific direction (down/up), and there are no net pressure differences of flow enforced in the simulations (as these are currently unknown in penetrating arteriole PVS), and the CSF oscillates in the PVS in a perfectly symmetrical fashion. When the concentration is 1 at the brain surface the concentration front will move downwards, if it is placed in the middle, the situation will be like in figure 4 where the solute is moved in both directions. Therefore, to perform the modeling as suggested, while technically not being a problem, would not add any information; it would be a perfect mirror image of the already presented results.

We have added the following sentence for clarity:

Page 7 line 160, page 8 line 183: *“The model does not enforce net pressure differences or net flow in the simulations.”*

6) Line 125: “there are cellular and non-cellular constituents within this compartment, such as smooth muscle cells, macrophages, fibroblasts, and extracellular matrix proteins^{25,26}, that likely hinder fluid flow.” This is logic since smaller space means higher resistance to flow. However, the simulations with the 25% and 50% PVS obstruction showed higher peak velocities than the 0%. How does the author explain this apparent contradiction?

The pressures, fluid flow and enhanced solute movement due to VLF and LF increases when obstruction such as SMC are added because the relative volume changes to the free unobstructed PVS increases when adding a fixed volume of cells and non-cellular constituents.

This is maybe better illustrated with an example: let's say the size of the PVS for a vessel is 100 units in baseline, and 70 units during a vasodilation. If we add a fixed volume of 50 units of hindrance (that does not permit flow) to this volume, and the vessel dilates, the respective volumes from the previous example would be 50 units and 20 units, respectively. In an even more extreme example (add 60–70 units) the PVS volume could go towards zero, in this simplified scenario meaning that all fluid is squeezed out. Since the fluid in the PVS is incompressible and needs to go somewhere when the PVS shrinks, the corresponding pressures and velocities must increase.

In other words, even though flow is hindered in some parts of the PVS (in our modeling added as the smooth muscle layer of the vessel wall), the velocities and flow are similarly increased in the areas that are open to flow. Although not entirely accurate in this context (because this is viscous flow), the situation is quite similar to what is described by the Bernoulli effect.

Reviewer #3 (Remarks to the Author):

Thank you to the authors for answering my questions and addressing my concerns. The revised manuscript is much improved and the title is more reflective of the data that is presented in the study. I would just request a few minor changes to the text:

We thank the reviewer for positive evaluation of our revised manuscript. The minor changes pointed out by the reviewer have now been amended in the revised manuscript.

Line 50: The previous paragraph has introduced the different models for flow along the PVS, thus one should rephrase “glymphatic” to “Flow along the PVS” at the beginning of the next paragraph.

The revised manuscript has been amended.

Line 256: Clarify that the CSF flow within the ventricles was shown to be reversed in the referenced study.

The revised manuscript has been amended.

Line 304: Please preface this sentence with “The model of” glymphatic waste clearance.

The revised manuscript has been amended.

REVIEWERS' COMMENTS

Reviewer #2 (Remarks to the Author):

The authors have responded well to my critique. It is a beautiful and important study. Ready for the publisher!